# CONSTRAINED FLOW OPTIMIZATION VIA SEQUENTIAL FINE-TUNING FOR MOLECULAR DESIGN

## ABSTRACT

Adapting generative foundation models, in particular diffusion and flow models, to optimize given reward functions (e.g., binding affinity) while satisfying constraints (e.g., molecular synthesizability) is fundamental for their adoption in real-world scientific discovery applications such as molecular design or protein engineering. While recent works have introduced scalable methods for reward-guided fine-tuning of such models via reinforcement learning and control schemes, it remains an open problem how to algorithmically trade-off reward maximization and constraint satisfaction in a reliable and predictable manner. Motivated by this challenge, we first present a rigorous framework for *Constrained Generative Optimization*, which brings an optimization viewpoint to the introduced adaptation problem and retrieves the relevant task of constrained generation as a sub-case. Then, we introduce **C**onstrained **F**low **O**ptimization (CFO), an algorithm that automatically and provably balances reward maximization and constraint satisfaction by reducing the original problem to progressive fine-tuning via established, scalable methods. We provide convergence guarantees for constrained generative optimization and constrained generation via CFO. Ultimately, we present an experimental evaluation of CFO on both synthetic, yet illustrative, settings, and a molecular design task optimizing quantum-mechanical properties.

## 1 INTRODUCTION

Recent advances in generative modeling, particularly the advent of diffusion (Ho et al., 2020; Song et al., 2021; 2022) and flow models (Lipman et al., 2023), have led to state-of-the-art performances in several fields such as image synthesis (Rombach et al., 2022), biology (Corso et al., 2023; Wohlwend et al., 2024), and chemistry (Hoogeboom et al., 2022). In particular, they have been applied for the design of protein structures (Wu et al., 2024), drug-like molecules (Dunn & Koes, 2024), and DNA sequences (Stark et al., 2024), among others. These generative models excel at capturing complex data distributions and generating realistic samples. However, approximately sampling from the data distribution is insufficient for most real-world discovery applications, where one typically wishes to generate candidates maximizing task-specific *rewards*, a problem recently denoted by *generative optimization* (De Santi et al., 2025b; Li et al., 2024). Examples of rewards of interest include binding affinity in drug discovery (Pantsar & Poso, 2018), or drug-likeness (Bickerton et al., 2012). To tackle the generative optimization problem, recent works have introduced scalable fine-tuning methods that adapt a pre-trained flow or diffusion model to maximize a given reward function under KL-regularization from the pre-trained model, via reinforcement learning (RL) or control theoretic methods (e.g., Domingo-Enrich et al., 2025; Uehara et al., 2024b; Tang, 2024).

**The importance of known constraints in generative optimization.** Many generative design and scientific discovery problems require generated samples to satisfy explicit, domain-specific constraints, e.g., bounded toxicity (Amorim et al., 2024), synthetic accessibility (Ertl & Schuffenhauer, 2009; Neeser et al., 2024), or biophysical plausibility of docking poses (Buttenschoen et al., 2024). Even though current fine-tuning schemes regularize toward a pre-trained model (Domingo-Enrich et al., 2025; Uehara et al., 2024b; Tang, 2024), which limits the distributional drift, they cannot certify hard constraints to be satisfied (Uehara et al., 2024a). This limitation arises because task-specific constraints may not be encoded in the original dataset or may be learned only imperfectly from finite training data. A naive approach to address such explicit constraints would be to include them as rewards, i.e., as another term in a manually weighted objective function.

However, this approach is unreliable in practice, as the appropriate weighting between rewards and constraints varies across tasks and training phases, and needs to be determined through inefficient trial and error. Furthermore, as optimization explores high-reward regions, the chosen weights can unexpectedly favor reward at the expense of constraint satisfaction, yielding samples with attractive rewards, which, however, violate the domain-specific constraints. Driven by these limitations of current flow adaptation methods for constraint satisfaction, we pose the following question:

*How can we fine-tune a pre-trained flow or diffusion model to reliably and predictably trade-off reward optimization and constraint satisfaction?*

**Our approach.** In this work, we aim to tackle this question by first introducing a formal framework for *Constrained Generative Optimization* (Sec. 3) via flow model fine-tuning, which entails adapting a pre-trained flow model to generate samples maximizing a reward function while satisfying arbitrary constraints. Moreover, the proposed formulation retrieves the relevant task of constrained generative modeling as the sub-case where the reward function is constant. Next, we introduce **C**onstrained **F**low **O**ptimization (CFO), a dual approach based on the augmented Lagrangian scheme (Birgin & Martínez, 2014) that turns the constrained objective into a sequence of ordinary generative optimization subproblems. At a high level, CFO alternates between two steps: solving a KL-regularized fine-tuning problem (Domingo-Enrich et al., 2025; Uehara et al., 2024b) to maximize an augmented reward function, and updating the parameters of the augmented reward using estimated constraint violations on generated samples (see Sec. 4). This progressively tunes the penalty on constraint violations, thereby avoiding the need for manual trade-off weight selection. CFO renders it possible to adapt a pre-trained flow model to maximize expected rewards while enforcing satisfaction of arbitrary constraints and preserving closeness to the pre-trained model. We provide guarantees that ensure constraint satisfaction under the realistic assumptions of an approximate solver, and that achieve reward maximization under a more idealized setting (Sec. 5). Finally, we evaluate CFO for both constrained generative modeling and constrained generative optimization problems, showcasing its performance in both visually interpretable illustrative settings and in molecular design tasks, showing constrained optimization of quantum mechanical properties. (Sec. 6).

**Our contributions.** To sum up, we present the following contributions:

- We propose a framework for constrained generative optimization via flow fine-tuning, capturing the practically relevant task of reward-guided adaptation under given constraints (Sec. 3).
- We introduce **C**onstrained **F**low **O**ptimization (CFO), an augmented Lagrangian-based method that provably tackles the introduced problem via progressive fine-tuning (Sec. 4).
- We provide guarantees for constrained generation and optimization via CFO under diverse oracle assumptions, by leveraging augmented Lagrangian theory for constrained optimization (Sec. 5).
- We demonstrate CFO's ability to trade-off reward maximization and constraint satisfaction in both visually interpretable settings and on high-dimensional molecular design tasks (Sec. 6).

## 2 BACKGROUND AND NOTATION

**Flow Models.** Flow-based generative models constitute a prominent class of approaches for transforming a simple base $p^{\text{base}}$ distribution (e.g., $p^{\text{base}} = \mathcal{N}(0, I)$) into a complex data distribution $p_{\text{data}}$ (Chen et al., 2018; Song et al., 2022; 2021; Lipman et al., 2023). Formally, a flow is a time-dependent map $\psi : [0, 1] \times \mathbb{R}^d \to \mathbb{R}^d$, where $\psi_t(x_0)$ denotes the position at time $t$ of a sample that started at $x_0$. The trajectory of $x_t$ $(:= \psi_t(x_0))$ is governed by a time-dependent velocity field $u : [0, 1] \times \mathbb{R}^d \to \mathbb{R}^d$ through the ordinary differential equation (ODE):

$$\frac{\mathrm{d}}{\mathrm{d}t}\psi_t(x_0) = u_t(\psi_t(x_0)), \quad \psi_0(x_0) = x_0. \tag{1}$$

A *generative* flow model defines a continuous-time Markov process $\{X_t\}_{t \in [0,1]}$, by sampling an initial value $X_0 \sim p^{\text{base}}$ and evolving it according to the flow map, $X_t = \psi_t(X_0)$. The terminal state $X_1 = \psi_1(X_0)$ is then required to follow the target distribution, i.e., $X_1 \sim p_{\text{data}}$. Equivalently, the flow induces a family of intermediate marginal densities $p_t$ describing the law of $X_t$ at each time $t \in [0, 1]$. We say that a velocity field $u$ generates the probability path $\{p_t\}_{t \in [0,1]}$ if the random variable $X_t = \psi_t(X_0)$ follows distribution $p_t$ for all $t < 1$. In practice, choosing $p^{\text{base}}$ simple (e.g., Gaussian) makes sampling tractable while $u_t$ provides the complexity needed to reach $p_{\text{data}}$.

**Flow Matching.** Flow Matching (Lipman et al., 2023) is a simulation-free algorithm to learn a vector field $u_\theta$, such that the induced marginal densities $p_t^{u_\theta}$ coincide with a prescribed probability path $\{p_t\}_{t\in[0,1]}$ and satisfying $p_0^{u_\theta} = p^{\text{base}}$ and $p_1^{u_\theta} = p_{\text{data}}$. Lipman et al. (2023) demonstrate that the Flow Matching and Conditional Flow Matching objectives share identical gradients, ensuring they converge to the same optimal vector field. In practice, this is achieved by introducing a reference flow and regressing the learned field $u_\theta(x_t, t)$ against the reference velocity:

$$\min_\theta \mathbb{E}_{t,p(x_0,x_1)} \left[ \left\| u_\theta(x_t, t) - \tfrac{d}{dt}\psi_t^{\text{ref}}(x) \right\|^2 \right]. \tag{2}$$

With an appropriate choice of the reference flow, specifically one that follows a diffusion trajectory, the Flow Matching framework recovers diffusion models as a particular case, showing that diffusion training objectives can be viewed as special instances of flow-based learning (Lipman et al., 2023; Domingo-Enrich et al., 2025). This formulation enables efficient training using only samples of $(t, x_0, x_1)$ and their corresponding reference velocities, without requiring expensive numerical integration. In practice, $u_\theta$ is parameterized by a neural network and sampling from $p_1^{u_\theta}$ ($\approx p_{\text{data}}$) is performed via simulating the ODE in Eq. 1.

**Reinforcement Learning in continuous-time.** Finite-horizon continuous-time reinforcement learning (RL) (Wang et al., 2020; Treven et al., 2023; Zhao et al., 2025) provides a principled framework for decision-making in dynamical systems and can be cast as an instance of optimal control. The state space is $\mathcal{X} := \mathbb{R}^d \times [0,1]$ and actions are taken from an action space $\mathcal{A}$. A (deterministic) policy $\pi : \mathcal{X} \to \mathcal{A}$ prescribes an action for each state $(x, t) \in \mathcal{X}$, yielding the dynamics:

$$\frac{d}{dt}\psi_t(x) = a_t(\psi_t(x)), \quad a_t = \pi(X_t, t), \quad X_0 \sim p^{\text{base}}. \tag{3}$$

The resulting process $\{X_t\}_{t\in[0,1]}$ induces a family of marginals $\{p_t^\pi\}_{t\in[0,1]}$. The aim is to optimize the expected performance, typically expressed through an integral reward accumulated along the trajectory and a terminal reward at $t = 1$ (Wang et al., 2020). In our setting, the reward over the trajectory is zero, and we focus solely on the terminal reward. We use RL notation to emphasize its generality and connection to standard practice, while noting that the setting coincides with deterministic optimal control since both the dynamics and the objective are known.

**Pre-trained Flow Models as RL Policy.** A pre-trained flow can be viewed as a feedback policy: at each time $t$ and state $x$, the velocity field $u^{\text{pre}}(x, t)$ prescribes the instantaneous action that determines how the system evolves. Defining $a_t = \pi^{\text{pre}}(X_t, t) := u^{\text{pre}}(X_t, t)$ for a policy $\pi^{\text{pre}} : \mathcal{X} \to \mathcal{A}$ (De Santi et al., 2025a), and substituting into Eq. 3, yields deterministic closed-loop dynamics. Starting from $X_0 \sim p_0$, rolling out $\pi^{\text{pre}}$ produces a trajectory $\{X_t\}_{t\in[0,1]}$ with induced marginals $\{p_t^{\pi^{\text{pre}}}\}_{t\in[0,1]}$. Intuitively, the policy selects at each moment the direction and speed that steer samples so that their distribution progressively matches the data, with the terminal marginal $p_1^{\text{pre}} := p_1^{\pi^{\text{pre}}}$ trained to approximate $p_{\text{data}}$. Viewing flow models through this policy lens not only unifies flow-based generation and control theory but also enables downstream fine-tuning as policy improvement with a terminal reward. For brevity, we refer to the pre-trained flow by its implicit policy $\pi^{\text{pre}}$.

## 3 CONSTRAINED GENERATIVE OPTIMIZATION VIA FLOW FINE-TUNING

In this work, we aim to fine-tune a pre-trained flow model $\pi^{\text{pre}}$ to obtain a new model $\pi^*$ inducing a process:

$$\frac{d}{dt}\psi_t(x) = a_t^{\text{fine}}(\psi_t(x)), \text{ with } a_t^{\text{fine}} = \pi^*(x_t, t). \tag{4}$$

such that its induced distribution $p_1^* := p_1^{\pi^*}$ maximizes the expected value of a property of interest, while satisfying arbitrary constraints and preserving prior information from $\pi^{\text{pre}}$. We denote this problem by *constrained generative optimization via fine-tuning*, illustrate in Figure 1 and defined as:

**Constrained Generative Optimization via Flow Fine-Tuning**

$$\arg\max_\pi \mathbb{E}_{x\sim p_1^\pi}[r(x)] - \alpha D_{KL}(p_1^\pi || p_1^{\text{pre}})$$

$$\text{s.t. } \mathbb{E}_{x\sim p_1^\pi}[c(x)] \leq B \tag{5}$$

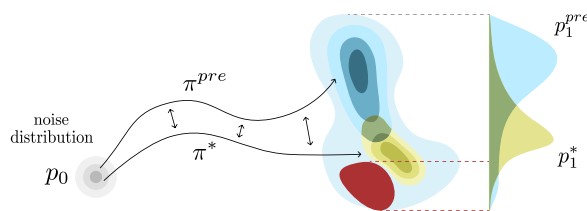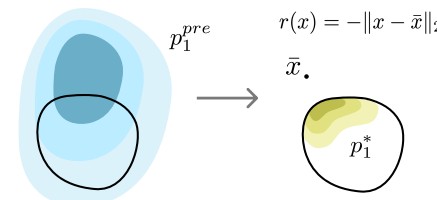

(a) Constrained generative optimization via fine-tuning problem. The red area has a high cost.

(b) Adaptation to low-cost area within black line.

Figure 1: (1a) Pre-trained and fine-tuned policies inducing densities $p_1^{pre}$ and optimal density $p_1^*$ w.r.t. reward $r$ increasing downwards and in red a high-cost area. (1b) Pre-trained model $p_1^{pre}$ adapts into $p_1^*$ to maximize $r$ and stay within the constraint region inside the black line.

Where $r : \mathcal{X} \to \mathbb{R}$ and $c : \mathcal{X} \to \mathbb{R}$ are a scalar reward and constraint function, $\alpha \in \mathbb{R}$ determines the KL-regularization strength, and $B \in \mathbb{R}$ is the upper bound on the constraint. Setting the reward term $r$ to be constant (e.g., $r = 0$) in Eq. 5, reduces the objective to a formulation of *constrained generation* as minimization of a KL divergence between the fine-tuned model density $p_1^\pi$ and the pre-trained model (i.e., $p_1^{\text{pre}}$), while satisfying the expected constraint bound in Eq. 5:

$$\arg\min_\pi \ \alpha D_{KL}(p_1^\pi || p_1^{\text{pre}}) \quad \text{s.t.} \quad \mathbb{E}_{x \sim p_1^\pi}[c(x)] \leq B \tag{6}$$

This problem has been studied before in (Chamon et al., 2025; Khalafi et al., 2025). A first approach to tackle Eq. 5 is to optimize a fixed-weight Lagrangian (Chamon et al., 2025; Zhang et al., 2025):

$$\max_\pi \ \mathcal{L}_\mu(\pi) \ = \ \mathbb{E}_{x \sim p_1^\pi}[r(x)] - \alpha D_{KL}(p_1^\pi || p_1^{\text{pre}}) - \mu \left( \mathbb{E}_{x \sim p_1^\pi}[c(x)] - B \right) \quad \text{s.t.} \quad \mu \geq 0 \tag{7}$$

Here, $\mu \in \mathbb{R}_{\geq 0}$ denotes the Lagrange multiplier that penalizes constraint violations. However, optimizing $\mathcal{L}_\mu$ with a fixed $\mu$ is unreliable for enforcing the constraint. First, feasibility (i.e., $\mathbb{E}_{x \sim p_1^\pi}[c(x)] \leq B$) is not guaranteed for any given $\mu$, unless it exceeds an unknown, problem-dependent threshold. Second, $\mu$ must be tuned by hand, and there is no guaranteed or monotone mapping from $\mu$ to the resulting violation, so trial-and-error often leads to either infeasible or overly conservative solutions. Finally, if $r$ is unbounded or approximate (e.g., a learned proxy reward function), maximizing $\mathcal{L}_\mu$ may shift probability mass toward high-reward regions, yielding invalid designs.

Toward overcoming such limitations, in the next section, we propose an algorithm that can provably tackle the constrained generative optimization problem introduced in Eq. 5 by progressively fine-tuning the initial pre-trained model via established methods (e.g., Domingo-Enrich et al., 2025).

## 4 CONSTRAINED FLOW OPTIMIZATION (CFO)

In the following, we introduce **Constrained Flow Optimization**, see Alg. 1, which addresses the *constrained generative optimization* problem as formulated in Eq. 5 by solving a sequence of unconstrained entropy-regularized fine-tuning subproblems, each with a different reward function computed via an augmented Lagrangian (AL) scheme (Rockafellar, 1976; Fortin, 1975; Birgin & Martínez, 2014). Intuitively, CFO tackles the problem by embedding the given constraint into an *augmented* reward via an adaptive penalty parameter, so that at each iteration, a standard entropy-regularized fine-tuning solver steers the model toward feasibility while improving reward.

**Overview of the Algorithm.** CFO (Alg. 1), takes as input a pre-trained model $\pi_{\text{pre}}$, a number of iterations $K$, a minimal Lagrange multiplier $\lambda_{\min} < 0$, an initial penalty parameter $\rho_{\text{init}} > 0$, a penalty growth rate $\eta \geq 1$, and a contraction value $0 < \tau < 1$. At each iteration $k$, CFO performs 5 main steps:

**Step 1:** An augmented objective $f_k$ (Eq. 9) is formed as the difference between the reward and a penalty term (Birgin & Martínez, 2014):

$$f_k(x) \ = \ r(x) - \frac{\rho_k}{2} \left[ \max\left( 0, c(x) - B - \frac{\lambda_k}{\rho_k} \right) \right]^2,$$

where the offset $\lambda_k / \rho_k \leq 0$ shifts the term toward the current expected constraint boundary.

---

**Algorithm 1** **C**onstrained **F**low **O**ptimization (CFO)

---

1: **Input:** $\pi_{\text{pre}}$: pre-trained model, $K$: number of iterations, $\lambda_{\min} < 0$: min. Lagrange multiplier, $\rho_{\text{init}} > 0$: initial penalty parameter, $\eta \geq 1$: growth rate, $0 < \tau < 1$: contraction value
2: **Init:** Set initial Lagrange multiplier $\lambda_1 = 0$ and penalty $\rho_1 = \rho_{\text{init}}$ parameters
3: **for** $k = 1, 2, \ldots, K$ **do**
4:     **Step 1:** Update fine-tuning AL objective:

$$f_k(x) := r(x) - \frac{\rho_k}{2} \left[ \max\left( 0, c(x) - B - \frac{\lambda_k}{\rho_k} \right) \right]^2 \tag{9}$$

5:     **Step 2:** Compute $\pi_k$ via fine-tuning:
$$\pi_k \leftarrow \text{FINETUNINGSOLVER}(f_k, \pi_{\text{pre}}) \tag{10}$$
6:     **Step 3:** Set the empirical constraint gap $G_k$ and contraction statistic $V_k$

$$G_k = \mathbb{E}_{x \sim p_1^{\pi_k}}[c(x)] - B \quad \text{and} \quad V_k = \min\{G_k, -\lambda_k/\rho_k\} \tag{11}$$

7:     **Step 4:** Compute Lagrange multiplier proposal:
$$\lambda_{k+1} \leftarrow \max\{\lambda_{\min}, \min\{0, \lambda_k - \rho_k G_k\}\} \tag{12}$$
8:     **Step 5:** Set the new penalty:

$$\rho_{k+1} = \begin{cases} \rho_k, & \text{if } k = 1 \text{ or } V_k \leq \tau V_{k-1}, \\ \eta \rho_k, & \text{otherwise} \end{cases} \tag{13}$$

9: **end for**
10: **Return:** $\pi_K$

---

**Step 2:** A FINETUNINGSOLVER (e.g., Domingo-Enrich et al., 2025) computes $\pi_k$ by solving a standard KL-regularized control (or RL) subproblem, with the current augmented objective $f_k$, namely:

$$\pi_k \in \arg\max_\pi \mathbb{E}_{x \sim p_1^\pi}[f_k(x)] - \alpha D_{KL}(p_1^\pi || p_1^{\text{pre}}), \tag{8}$$

For completeness, we report a detailed implementation of this *oracle* step in Appendix A.

**Step 3:** CFO computes a Monte Carlo estimate of the constraint $c$ under the current policy $\pi_k$ (see Eq. 11), and subtracts the user-defined bound B, thus obtaining the *empirical constraint gap* $G_k$. Then, it computes a *contraction statistic* $V_k$, which measures the current progress toward feasibility by comparing the recent estimate $G_k$ of the constraint gap with the $\lambda_k/\rho_k \leq 0$ offset term.

**Step 4:** Then, CFO uses the empirical constraint gap $G_k$ to apply a projected dual update to the Lagrange multiplier (see Eq. 12). If $G_k > 0$ (i.e., the constraint is violated), and the multiplier $\lambda_{k+1}$ is decreased. This shifts the penalty toward the new current expected constraint boundary (i.e., $G_k - \lambda_k/\rho_k$). Instead, if $G_k < 0$ (i.e., the constraint is fulfilled), then the Lagrange multiplier $\lambda_k$ is increased toward 0.

**Step 5:** The contraction statistic $V_K$ (see Eq. 11) assesses progress toward feasibility. If $V_k$ does not contract sufficiently, i.e., $V_k > \tau V_{k-1}$, where $\tau$ is a user-defined contraction rate, then CFO infers that the penalty is not sufficiently high and thus increases it by a multiplicative factor $\eta$. Instead, if $V_k$ is contracting, $\rho$ is kept fixed, as shown in Eq. 13. Ultimately, CFO returns the fine-tuned policy $\pi_K$.

A discussion on hyperparameters can be found in Appendix D. Nevertheless, it is a priori unclear whether CFO is guaranteed to solve the constrained generative optimization problem in Eq. 5. In the next section, we provide an affirmative answer by showing that under oracle assumptions, CFO achieves reward optimality and arbitrary constraint satisfaction.

## 5 CONSTRAINED GENERATIVE OPTIMIZATION GUARANTEES

Before presenting the convergence properties of CFO, we first establish a mild and realistic assumption on the FINETUNINGSOLVER used in Alg. 1, which formalizes the approximate nature of its optimization steps and serves as the foundation for the theoretical guarantees that follow.

**Assumption 5.1** (Approx. Solver). At every iteration $k$, the solver outputs a policy $\pi_k$ satisfying:

$$L_{\rho_k}(\pi_k, \lambda_k) \geq L_{\rho_k}(\pi, \lambda_k) - \varepsilon_k, \quad \forall \pi \tag{14}$$

where $L_{\rho_k}(\pi_k, \lambda_k) = \mathbb{E}_{x \sim p_1^\pi}[f_k(x)] - \alpha D_{KL}(p_1^\pi || p_1^{\text{pre}})$ and the sequence $\{\varepsilon_k\} \subseteq \mathbb{R}_+$ is bounded.

This assumption captures the approximate nature of practical fine-tuning or optimization oracles, it is standard in augmented Lagrangian (AL) frameworks and has been adopted in recent works (e.g., De Santi et al., 2025a). The key requirement is that the approximation error remains bounded.

To keep the notation simple, we will define the infeasibility of a policy $\pi$ as:

$$G(\pi) = \mathbb{E}_{x \sim p_1^\pi}[c(x)] - B. \qquad (15)$$

If the infeasibility $G(\pi)$ of a given policy is positive, the policy is infeasible, i.e., its average constraint is larger than the permissible bound. If $G(\pi)$ is negative, the policy is feasible and thus fulfills the constraint. Using Assumption 5.1 and Eq. 15, we state our main convergence results for CFO. The proofs are in Appendix E and draw on the analysis developed by Birgin & Martínez (2014).

**Theorem 5.2** (Feasibility of CFO). *Let $\{\pi_k\}$ be a sequence generated by Alg. 1 under Assumption 5.1 on the FINETUNINGSOLVER. Let $\bar\pi$ be a limit of the sequence $\{\pi_k\}$. Then, we have:*

$$\langle G(\bar\pi) \rangle_+ \leq \langle G(\pi) \rangle_+ \quad \forall \pi \qquad (16)$$

*where $G(\pi)$ is defined in Eq. 15 and $\langle \cdot \rangle_+ := \max\{0, \cdot\}$*

Concretely, Theorem 5.2 states that CFO returns a policy that minimizes the introduced infeasibility measure (Eq. 15). Thus, finding either a feasible policy or a policy that minimizes the constraint violations as far as possible.

**Corollary 5.3** (Feasibility of the Limiting Policy). *Under the same conditions as Theorem 5.2, if the underlying problem admits a feasible policy, then the limiting policy $\bar\pi$ is feasible, i.e., it satisfies the constraint (i.e., $G(\bar\pi) \leq 0$).*

Theorem 5.2 and Corollary 5.3 establish constraint satisfiability of CFO but do not yet show optimality of the returned policy. To achieve optimality, CFO requires a stronger assumption on the FINETUNINGSOLVER, namely that the approximation error vanishes asymptotically, i.e., $\varepsilon_k \to 0$.

**Theorem 5.4** (Optimality of CFO). *Let $\{\pi_k\}$ be the sequence generated by Alg. 1 under Assumption 5.1 with $\lim_{k\to\infty} \varepsilon_k = 0$ (in Eq. 14). Let $\bar\pi$ be a limit of the sequence $\{\pi_k\}$. Suppose that the problem in Eq. 5 is feasible, i.e., $\langle G(\bar\pi) \rangle_+ = 0$. Then, the limiting policy $\bar\pi$ is a global maximizer.*

Although having access to a FINETUNINGSOLVER achieving $\varepsilon_k \to 0$ exactly is rarely possible in practice, for our experiments (Sec. 6), we use Adjoint Matching (Domingo-Enrich et al., 2025). If the FINETUNINGSOLVER archives such a tight bound highly depends on the application and the complexity of the reward function. Our experiments demonstrate that CFO can achieve near-optimal reward performance while consistently respecting the constraint, even with bounded error.

The convergence guarantees of CFO do not rely on $r$ or $c$ being differentiable. Any further assumptions stem from the FINETUNINGSOLVER. Hence, using a gradient-free FINETUNINGSOLVER extends the applicability of CFO to problems where $r$ and $c$ are accessed purely through function evaluations.

## 6 EXPERIMENTAL EVALUATION

We demonstrate the ability of **C**onstrained **F**low **O**ptimization (Alg. 1) to solve the *constrained generative optimization* problem (see Eq. 5) on both low-dimensional illustrative settings, and on molecular design tasks. In particular, we evaluate: (i) the performance of CFO to solve Problem 5 given visually interpretable reward and constraint functions, also for (ii) the sub-case of constrained generation, recovered via a constant reward (see Eq. 6). We further show that (iii) CFO scales to high-dimensional molecular design tasks, and that (iv) it shows promising performances even with an approximate FINETUNINGSOLVER, or when run with a limited number of iterations $K$.

**CFO reliably solves constrained generative optimization low-dimensional tasks.** We first evaluate CFO's ability to solve the *constrained generative optimization* problem (see Eq. 5) on a visually interpretable setting, where $p_1^{\text{pre}}$ is a mixture of two non-overlapping Gaussians as shown in Figure 2a, enabling direct visualization of constraint satisfaction during fine-tuning. In this setting, the reward $r$ is the negative squared distance to the white cross in Figures 2a-2c (see color-coding

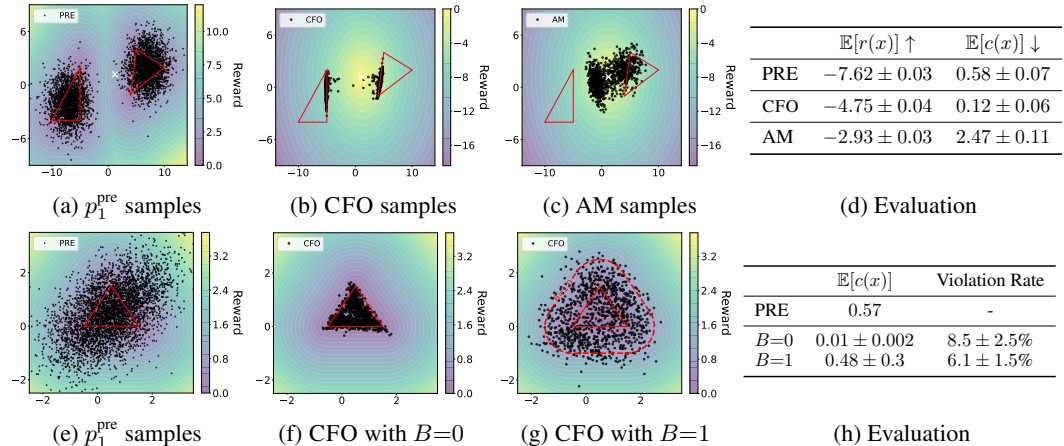

(a) $p_1^{\text{pre}}$ samples     (b) CFO samples     (c) AM samples     (d) Evaluation

(e) $p_1^{\text{pre}}$ samples     (f) CFO with $B{=}0$     (g) CFO with $B{=}1$     (h) Evaluation

Figure 2: 2a and 2e: Samples from the pre-trained models $p_1^{\text{pre}}$ and the constraint-free area is inside the red triangles. **(top) Constrained Generative Optimization:** Samples from fine-tuned models via CFO (2b) and Adjoint Matching (AM) (Domingo-Enrich et al., 2025) (2c). **(bottom) Constrained generation:** Samples from the fine-tuned model via CFO with $B{=}0$ (2f) and $B{=}1$ (2g). Tables showing numerical results for the respective rows (2d and 2h).

in Figure 2b and 2c.) The constraint $c$ is zero within the red triangles in Figures 2a-2c, and increases linearly outside (see color-coding in Figure 2a). As shown in Figure 2b, CFO, run with $K = 20$, and $\rho_{\text{init}} = 0.5$, steers the pre-trained flow model such that its induced density $p^*$ is located predominantly within the valid regions (i.e., red triangles) where the constraint is fulfilled, while simultaneously optimizing the reward by moving samples toward the inner boundaries of both triangles. CFO increases the mean reward from $-7.62$ to $-4.75$ compared to the base model, while it reduces estimated constraint violations from $0.58$ to $0.12$, as reported in Table 2d. The minor residual violations of CFO, which one can notice e.g., in Figure 2b, are likely due to Monte Carlo approximation errors during finetuning. In contrast to CFO, Adjoint Matching (Domingo-Enrich et al., 2025), a well-established reward-guided fine-tuning scheme, which does not take into account any constraint, raises the expected reward to $-2.93$, but significantly degrades the models ability to satisfy the given constraints, increasing constraint violations from $0.58$ to $2.47$ (see Figure 2c).

**Constant reward recovers Constrained Generation.** To illustrate the constrained generation (see Eq. 6) capabilities, we consider a correlated Gaussian base density $p_1^{\text{pre}}$, visualized in Figure 2e, and a constraint $c$ penalizing samples outside the red central triangle (see Figure 2e). In the following, we vary the bound $B \in \{0.0, 1.0\}$ (see Eq. 5) to obtain diverse flow models inducing fine-tuned distributions $p^*$. As shown in Figures 2f–2g, by increasing $B$, the resulting densities visibly expand beyond the zero constraint region, illustrating the relaxation of constraint enforcement. Quantitatively, the selected degree of permissible violation (i.e., the value of B), is reflected in the mean constraint violations incurred by the respective flow models, obtained by running CFO with $K = 20$, and $\rho_{\text{init}} = 0.5$. As shown in Table 2h, while setting $B = 1$ leads to expected constraint value of $0.01$, choosing $B = 1.0$ renders CFO less restrictive, inducing a policy $\pi^*$ with a mean constraint of $0.48$. While the base model exhibits $\mathbb{E}_{p_1^{\text{pre}}}[c(x)] = 0.57$, the violation decreases to $0.48$ under $B = 1.0$ and further to $0.01$ under $B = 0.0$. These results illustrate how the choice of $B$ controls tolerance to constraint violations, offering a mechanism to adapt CFO to domain-specific requirements.

**CFO scales to high-dimensional molecular design tasks.** To demonstrate the practical relevance of CFO in high-dimensional settings, we apply CFO to a molecular design, where satisfying constraints is critical. Specifically, we adapt FlowMol (Dunn & Koes, 2024), a flow model pre-trained on GEOM Drugs (Axelrod & Gómez-Bombarelli, 2022), and maximize the dipole moment (Minkin et al., 1970) as reward while ensuring constraint fulfillment. As constraints, we impose an upper bound on the total xTB energy (i.e., $-80$ Ha), to be used as a proxy for chemical stability. Further details on the constraint and reward functions employed are provided in Appendix B. Both functions are computed via GNN-based predictors (see Appendix B) trained on GFN2-xTB (Bannwarth et al., 2019). While we employ differentiable rewards and constraints, this is rather a requirement of the specific FINETUNINGSOLVER we use in our implementation, namely Adjoint

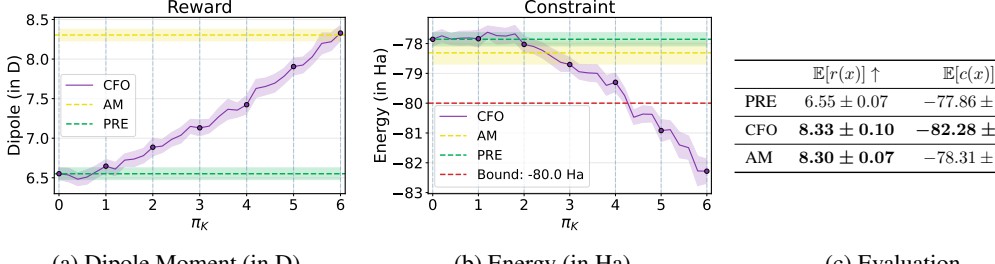

(a) Dipole Moment (in D)    (b) Energy (in Ha)    (c) Evaluation

Figure 3: Energy-constrained dipole moment maximization of FlowMol (Dunn & Koes, 2024) on GEOM Drugs (Axelrod & Gómez-Bombarelli, 2022). (3a-3b): Evolution of the constraint and reward during CFO fine-tuning with ($K = 6$, $N = 10$) in comparison to AM (Domingo-Enrich et al., 2025), which is run for $N = 60$ steps, and we show the final iterate. 3c: Numeric Evaluation of CFO ($K = 6$, $N = 10$) and AM ($N = 60$) on the molecular design task (best are bold). For all figures, report the mean and $95\%$ CI (32 seeds); vertical lines indicate parameter updates.

Matching (Domingo-Enrich et al., 2025), rather than a need of our method, which is compatible with non-differentiable reward and constraint functions (see Sec. 5).

In Figure 3, we show the performance of CFO for the energy-constrained dipole moment max-imization molecular design task. The optimal policy $\pi^*$ computed by CFO ($K = 6, N = 10$) increases the dipole moment from 6.55 Debye of the pre-trained model to 8.33 Debye (see Figure 3a). Simultaneously, $\pi^*$ shifts the flow model density to generate predominantly low-energy samples, effectively achieving an expected energy of $-82.28$ Ha, thus satisfying the upper bound B of $-80$ Ha. In Figure 4, we present drug-like samples from the fine-tuned model, together with their ground-truth reward and constraint values. For reference, running Adjoint Matching ($N = 60$) (Domingo-Enrich et al., 2025) purely for reward maximization, without enforcing the constraint, achieves a similar reward of 8.30 Debye, yet results in an expected constraint of $-78.31$ Ha, thus not fulfilling the constraint (see Table 3c). Appendix B shows that GNN predictors are accurate throughout the optimization, with ground truth values of reward and constraint being optimized to the same extent.

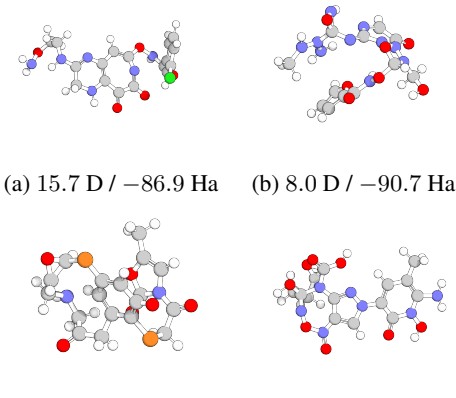

(a) 15.7 D / $-86.9$ Ha    (b) 8.0 D / $-90.7$ Ha

(c) 9.1 D / $-83.4$ Ha    (d) 12.5 D / $-93.8$ Ha

Figure 4: Drug-like molecules sampled from the fine-tuned model, together with ground-truth dipole moments (D) and energies (Ha).

We observe that optimization of the molecular properties leads to a decrease in the fraction of valid generated molecules (from $35\%$ to $9\%$ for CFO and $4\%$ for AM). This is expected, as validity is not directly enforced but only implicitly learned from the training distribution. The fine-tuning shifts the model toward less represented regions of chemical space, where this implicit notion of validity becomes less reliable. In Appendix B, we discuss how base model improvements and differentiable geometry relaxation could increase the validity of generated molecules for downstream applications.

To contextualize the effects of reward-guided fine-tuning, we report standard molecular statistics for models fine-tuned with CFO and AM. Although these graph-based metrics are not optimization targets, they show how molecular properties shift when the model is steered toward high dipole moments under energy constraints. As reported in Table 5c, both CFO and AM perform similarly, e.g., the QED score, where CFO achieves 0.38 and AM 0.37, coming from 0.45 by the base model.

Moreover, to illustrate CFO's versatility across different constraint formulations, we replace the energetic constraint with a molecular validity criterion based on `PoseBusters` (Buttenschoen et al., 2024), the results of which are presented in Appendix C. Beyond GNN-based surrogates, we also show the performance of CFO on ground-truth rewards and constraints from a differentiable

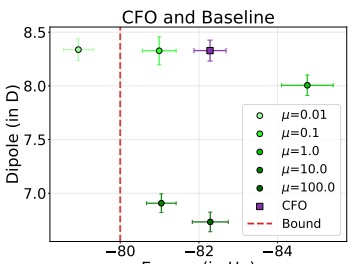

(a) CFO and fixed $\mu$-Baseline

|  | $\mathbb{E}[r(x)] \uparrow$ | $\mathbb{E}[c(x)] \downarrow$ |
|---|---|---|
| PRE | $6.55 \pm 0.07$ | $-77.86 \pm 0.22$ |
| CFO | $\mathbf{8.33 \pm 0.10}$ | $-82.28 \pm 0.41$ |
| $\mu = 0.01$ | $\mathbf{8.34 \pm 0.10}$ | $-78.94 \pm 0.38$ |
| $\mu = 0.1$ | $\mathbf{8.33 \pm 0.13}$ | $-80.99 \pm 0.43$ |
| $\mu = 1.0$ | $8.01 \pm 0.10$ | $\mathbf{-84.75 \pm 0.66}$ |
| $\mu = 10.0$ | $6.91 \pm 0.09$ | $-81.05 \pm 0.38$ |
| $\mu = 100.0$ | $6.73 \pm 0.09$ | $-82.29 \pm 0.46$ |

(b) Evaluation

| Feature | PRE | CFO | AM |
|---|---|---|---|
| Validity (in %, ↑) | 34 | 9 | 4 |
| QED (↑) | 0.45 | 0.38 | 0.37 |
| Lipinski (in %, ↑) | 90 | 76 | 76 |
| logP (↑) | 0.9 | $-0.89$ | $-0.55$ |
| Murcko scaffolds | 344 | 84 | 34 |

(c) Molecular Statistic

Figure 5: (5a): Pareto plot of the molecular design task, comparing CFO ($K = 6$, $N = 10$) against multiple fixed-$\mu$ baselines (see Eq. 7) ran with AM ($N = 60$). 5b: Numeric Evaluation of (5a)(best are bold, 32 seeds). (5c): Molecular statistics. Validity: RDKit-based validity checker (Landrum, 2025), QED (Ertl & Schuffenhauer, 2009), Lipinski: percentage of valid molecules that fulfill all criteria of Lipinski's rule of 5 (Lipinski, 2004), logP: average logP values of valid molecules, Murcko scaffolds: number of Murcko scaffolds in valid molecules (out of 1000 molecules)

simulator, namely dxTB (Friede et al., 2024). Similar to previous experiments, we find that CFO increases the reward while fulfilling the given constraints (Appendix C).

**CFO outperforms a fixed-$\mu$ baseline.** Comparing CFO against a fixed-$\mu$ baseline (Eq. 7) empirically validates the observation outlined in Sec. 3: manually tuning $\mu$ is unreliable (see Figure 5a and Table 5b). When $\mu$ is set too small (e.g., $\mu = 0.01$), the baseline attains a high reward (8.34 Debye) but exhibits substantial constraint violation ($-78.94$ Ha). Conversely, when $\mu$ is large (e.g., $\mu \geq 1.0$), the constraint is satisfied, but the reward drops significantly (8.01 Debye for $\mu = 1.0$), falling short of the performance achieved by CFO. These findings show that the online parameter adaptation in CFO provides a more robust mechanism for balancing reward maximization and constraint satisfaction. We further find that CFO remains robust across a range of parameter choices. An ablation study is provided in Appendix D.

**CFO can run with approximate fine-tuning oracles and a limited number of iterations $K$.** While CFO has $K$ outer iterations, typical fine-tuning solvers (Domingo-Enrich et al., 2025; Uehara et al., 2024c; Tang, 2024) require $N$ steps to compute the optimal iterates. This makes CFO a double loop algorithm. But in practice, we run CFO under a fixed solver-step budget of $M = K \cdot N$ for all experiments, thus keeping the total compute constant. This leads to a trade-off between the exactness of the *inner* solver and the *outer* dual updates. Increasing $K$ reallocates budget from a more exact inner solver to more frequent updates of the Lagrange parameters, effectively making the FINETUNINGSOLVER less precise at every *outer* step.

To show that CFO can effectively work with an approximate fine-tuning oracle, we probe the setting shown in Figures 2a–2c. Empirically, under a fixed budget of $M = 6000$, varying $K$ reveals a clear trade-off between constraint satisfaction and reward. When using very few dual updates ($K = 3$), the inner solver remains highly accurate ($N = 2000$), resulting in high reward but also high expected constraint violations (0.40). Conversely, using $K = 100$ produces very frequent dual updates, but makes the inner solver approximate

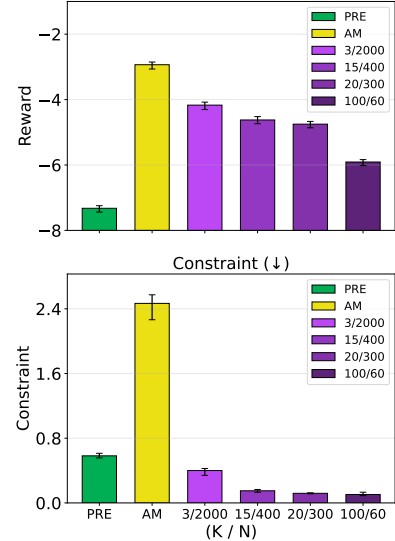

Figure 6: Reward and constraint for different values of (K/N)

($N = 60$), which almost eliminates the expected constraint violations (0.10) but substantially decreases the reward ($-5.91$). An intermediate configuration ($K = 20$) achieves a favorable balance, yielding both low constraint violation (0.12) and high reward (2.47), as shown in Figure 6. Thus CFO effectively acts as a fixed-budget allocator, balancing solver precision and update frequency, where moderately inexact inner solvers allow more dual updates, and thus better constraint satisfac-

tion. This implies that from a practical standpoint, the computational cost of CFO is comparable to that of standard fine-tuning schemes such as AM (Domingo-Enrich et al., 2025).

Importantly, this observation also holds for the molecular design task in Figure 3. CFO ($K = 6$, $N = 10$) and AM ($N = 60$) have comparable computational cost, as both perform 60 gradient steps. Concretely, CFO. has a total runtime of 37.18 min and compares well to the runtime of AM with 35.35 min. This 5% increase arises from the extra sampling and constraint evaluation performed in Step 3 of Alg. 1. Thus demonstrating that CFO can operate effectively in high-dimensional domains even with an approximate oracle.

## 7   RELATED WORK

**Control-based fine-tuning of flow and diffusion models.** Recent works have tackled fine-tuning of diffusion and flow models to maximize rewards under KL regularization as an entropy-regularized optimal control problem (e.g., Uehara et al., 2024b; Tang, 2024; Uehara et al., 2024c; Domingo-Enrich et al., 2025). Such methods have been successfully applied to real-world domains such as image generation (Domingo-Enrich et al., 2025), molecular design (Uehara et al., 2024c), or protein engineering (Uehara et al., 2024c). These methods have also been adopted as subroutines to tackle settings beyond reward maximization, such as manifold exploration (De Santi et al., 2025a) or optimization of distributional objectives (De Santi et al., 2025b). CFO extends fine-tuning methods for reward maximization to leverage known constraint functions and can be straightforwardly used as a plug-in oracle in more complex settings (e.g.,, exploration and distributional fine-tuning).

**Constrained Generative Modeling and Optimization.** Most prior work addresses constraint-aware generative modeling, developing tools for handling linear (Graikos et al., 2025), differentiable (Khalafi et al., 2024), and black-box (Kong et al., 2024) constraints. Enforcement spans training-time dual/penalty formulations (Khalafi et al., 2024) and inference-time strategies such as reward-weighted denoising for non-differentiable objectives (Kong et al., 2024) and classifier or classifier-free guidance for differentiable surrogates (Dhariwal & Nichol, 2021; Ho & Salimans, 2022). These techniques have been applied in domains such as molecular design (Kong et al., 2024) and constrained planning (Ma et al., 2025). The closest work to ours is arguably (Khalafi et al., 2024), with the main difference that our setting is for post-training, i.e., at fine-tuning time, constrained generative optimization rather than a training-time scheme enforcing given constraints.

**Augmented Lagrangian and Dual Methods in Constrained Sampling.** Augmented Lagrangian and dual formulations turn equality and inequality constraints into auxiliary updates that run with the sampler, enabling draws from unnormalized targets while enforcing feasibility either per-sample or in expectation (Khalafi et al., 2025; Blanke et al., 2025; Chamon et al., 2025). For example, in planning and control, Zhang et al. (2025) employ an augmented Lagrangian method to steer diffusion rollouts toward time-varying safety sets without requiring retraining of the base model. Dual schemes similarly maintain physical invariants during sampling or data assimilation while still retaining sufficient exploration of feasible states (Blanke et al., 2025). In addition to constraint generation or sampling, CFO also performs reward-driven optimization under the augmented formulation.

## 8   CONCLUSION

This work tackles the problem of *constrained generative optimization* via fine-tuning of pre-trained flow and diffusion models, a relevant and challenging task in discovery applications such as drug discovery. After proposing a constrained optimization formulation of the problem, we introduced **C**onstrained **F**low **O**ptimization, a method that transforms the constrained objective into a sequence of fine-tuning steps, and provides feasibility and optimality guarantees. Empirical results on both illustrative settings and molecular design tasks demonstrate the ability of CFO to steer pre-trained flow models toward high-reward regions while satisfying the given constraints. Promising directions include adding zero-order oracles to CFO beyond the current first-order choice, developing inference-time constraint handling rather than fine-tuning, and testing on protein engineering tasks.

## 9 REPRODUCIBILITY STATEMENT

We have taken several steps to ensure the reproducibility of our results. To facilitate replication, we provide a complete description and pseudocode of the algorithm in the main text 4, along with pseudocode of the FINETUNINGSOLVER in Appendix A. All experimental settings, hyperparameters, and implementation details necessary to reproduce our results are documented in Appendix B. For the data and models, we use publicly available weights and code. For the 2D experiments, we describe the data-generating process and models in the Appendix B-C. For theoretical components, we clearly state all assumptions and provide complete derivations of key results in Section 5 and Appendix E.

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

## A  IMPLEMENTATION OF FINETUNINGSOLVER- ADJOINT MATCHING (DOMINGO-ENRICH ET AL., 2025)

To ensure completeness, below we provide pseudocode for one concrete realization of a FINETUN-INGSOLVER as in Eq. 10. We describe exactly the version employed in Sec. 6, which builds on the Adjoint Matching framework (Domingo-Enrich et al., 2025), casting linear fine-tuning as a stochastic optimal control problem and tackling it via regression.

Let $u^{\text{pre}}$ be the initial, pre-trained vector field, and $u^{\text{finetuned}}$ its fine-tuned counterpart. We also use $\bar{\alpha}$ to refer to the accumulated noise schedule from Ho et al. (2020), effectively following the flow models notation introduced by Adjoint Matching (Domingo-Enrich et al., 2025, Sec. 5.2). The full procedure is in Alg. 2.

---

**Algorithm 2** FINETUNINGSOLVER- Adjoint Matching (Domingo-Enrich et al., 2025)

---

1: **Input:** $N$ : number of iterations, $u^k$ : current finetuned flow vector field, $u^{\text{pre}}$ : pre-trained flow vector field, $\alpha$ regularization coefficient (Eq. 5), $\nabla f$: objective function gradient, $m$ batch size, $h$ step size
2: **Init:** $u^{\text{finetuned}} := u^k$ with parameter $\theta$
3: **for** $n = 0, 1, 2, \ldots, N-1$ **do**
4:     Sample $m$ trajectories $\{X_t\}_{0 \leq t \leq 1}$ via a memoryless noise schedule $\sigma(t)$ (Domingo-Enrich et al., 2025), e.g.,

$$\text{sample } \varepsilon_t \sim \mathcal{N}(0, I), \ X_0 \sim \mathcal{N}(0, I), \text{ then:} \tag{17}$$

$$X_{t+h} = X_t + h \left( 2u_\theta^{\text{finetuned}}(X_t, t) - \frac{\bar{\alpha}_t}{\alpha_t} X_t \right) + \sqrt{h}\sigma(t)\varepsilon_t \tag{18}$$

5:     Use objective function gradient:

$$\tilde{a}_1 = -\frac{1}{\alpha} \nabla_{X_1} f(X_1)$$

6:     For each trajectory, solve the lean adjoint ODE, (Domingo-Enrich et al., 2025, Eq. 38-39), from $t = 1$ to 0:

$$\tilde{a}_{t-h} = \tilde{a}_t + h\tilde{a}_t^\top \nabla_{X_t} \left( 2u^{\text{pre}}(X_t, t) - \frac{\bar{\alpha}_t}{\alpha_t} X_t \right) \tag{19}$$

7:     Where $X_t$ and $\tilde{a}_t$ are computed without gradients, i.e., $X_t = \texttt{stopgrad}(X_t), \tilde{a}_t = \texttt{stopgrad}(\tilde{a}_t)$. For each trajectory, compute the Adjoint Matching objective (Domingo-Enrich et al., 2025, Eq. 37):

$$\mathcal{L}_\theta = \sum_{t \in \{0, h, \ldots, 1-h\}} \left\| \frac{2}{\sigma(t)} \left( u_\theta^{\text{finetuned}}(X_t, t) - u^{\text{pre}}(X_t, t) \right) + \sigma(t)\tilde{a}_t \right\|^2 \tag{20}$$

8:     Compute the gradient $\nabla_\theta \mathcal{L}(\theta)$ and update $\theta$.
9: **end for**
10: **Output:** Fine-tuned flow vector field $u_\theta^{\text{finetuned}}$

---

For further implementation details, we refer to Domingo-Enrich et al. (2025, Appendix G).

## B  FURTHER EXPERIMENTS AND DETAILS - ILLUSTRATIVE EXAMPLES

**Reward-only rejection sampling.** We also compare against a simple rejection-sampling baseline, complementary to the fixed-$\mu$ baseline in Eq. 7. We fine-tune a policy purely on the reward signal using Adjoint Matching and then enforce feasibility only by rejecting samples that violate the constraint. On the example in Figure 2c, this reward-only policy attains a constraint satisfaction rate of 13.40%, compared to 84.40% for the policy fine-tuned with CFO, e.g., accounting for the constraint during fine-tuning. Inspecting the samples further reveals that (1) violations under CFO occur predominantly near the constraint boundary, and (2) rejection sampling is ineffective when the reward optimum and the constraint region are poorly aligned.

**Details for visually interpretable settings (Figure 2).** The Mixture of Gaussians (Figure 2a) is generated by

$$p(x) = \frac{1}{2}\left(\mathcal{N}\left(x \mid \begin{bmatrix} -7 \\ -2 \end{bmatrix}, \Sigma\right) + \mathcal{N}\left(x \mid \begin{bmatrix} 2 \\ 7 \end{bmatrix}, \Sigma\right)\right), \quad \text{with } \Sigma = \begin{bmatrix} 3 & 0 \\ 0 & 3 \end{bmatrix},$$

We sample $20k$ points (80/20 train/validation split) and train a MLP with 3 hidden layers, each with 256 nodes, for the vector field $v$. The same setting is used for the experiment on the correlated Gaussian (Figure 2e), with:

$$p(x) = \mathcal{N}\left(x \mid \begin{bmatrix} 0.5 \\ 0.5 \end{bmatrix}, \begin{bmatrix} 1 & 0.5 \\ 0.5 & 1 \end{bmatrix}\right)$$

The constraint triangles have the following vertices:

1. **MoG:**
$$\triangle^I : \left(\begin{bmatrix} -10 \\ -4 \end{bmatrix}, \begin{bmatrix} -5 \\ -4 \end{bmatrix} \begin{bmatrix} -5 \\ 2 \end{bmatrix}\right) \quad \text{and} \quad \triangle^{II} : \left(\begin{bmatrix} 4 \\ -1 \end{bmatrix}, \begin{bmatrix} 10 \\ 2 \end{bmatrix}, \begin{bmatrix} 5 \\ 4 \end{bmatrix}\right)$$

2. **Correlated Gaussian:**
$$\triangle : \left(\begin{bmatrix} -1 \\ -0.5 \end{bmatrix}, \begin{bmatrix} 1 \\ -0.5 \end{bmatrix}, \begin{bmatrix} 0 \\ 1 \end{bmatrix}\right)$$

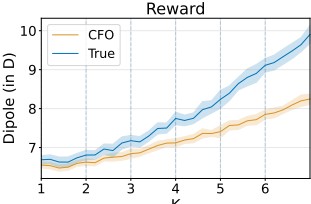

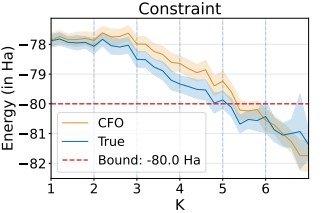

| **MD** | $\mathbb{E}[r(x)] \uparrow$ | $\mathbb{E}[c(x)] \downarrow$ |
|---|---|---|
| True PRE | $6.68 \pm 0.28$ | $-77.71 \pm 0.55$ |
| CFO | $8.37 \pm 0.26$ | $-81.68 \pm 0.71$ |
| `xtb` | $10.31 \pm 0.46$ | $-81.42 \pm 0.88$ |

(a) Dipole Moment (in D)     (b) Energy (in Ha)     (c) Evaluation

Figure 7: Energy-Constrained Dipole Moment Maximization for Molecular Design (MD) (7a-7b): Evolution of the constraint and reward during CFO compared to the true `xtb` Value. 7c: Numeric Comparison between of CFO and `xtb`.

## C   FURTHER RESULTS ON MOLECULAR DESIGN EXPERIMENTS

**Molecular Design.** For the molecular design task, we fine-tune FlowMol (Dunn & Koes, 2024). FlowMol models the molecules as graphs $g = (X, A, C, E)$, where $X = \{x_i\}_{i=1}^N \in \mathbb{R}^{N \times 3}$ is the atom position matrix, $A = \{a_i\}_{i=1}^N \in \mathbb{R}^{N \times n_a}$ are the atom types, $C = \{c_i\}_{i=1}^N \in \mathbb{R}^{N \times n_c}$ denote the formal charges, and $E = \{e_{ij} \mid \forall i, j \in [N] | i \neq j\} \in \mathbb{R}^{N^2 - N \times n_e}$ the bond order matrix. Where $n_a$, $n_c$, and $n_e$ are the number of possible atom types, charges, and bond orders, these are categorical variables represented by one-hot vectors. We refer to (Dunn & Koes, 2024) for the sampling of categorical and initial values. We use Gaussian sampling for the experiments in the main text on GEOM-Drugs and CTMC for the experiments on QM9.

**GNN Details and Generalization.** To verify that optimization targets the intended physical objective rather than exploiting the surrogate, we evaluate the ground-truth `xTB` values for every molecule sampled during the execution of CFO and compare their properties to the GNN predictions. For the energy (used as a constraint), surrogate predictions are essentially indistinguishable from `xTB`, indicating faithful approximation within the explored region. For the dipole moment (the maximization target), the surrogate systematically underestimates the true xTB values by $10\%$, yet the two remain strongly correlated and move in lockstep throughout the fine-tuning. Consequently, improvements under the surrogate translate to larger gains under `xTB`. Overall, these checks indicate that CFO does not exploit model artifacts and remains within the training distribution.

**Additional Results with Exact Rewards and Constraints using `dxtb`.** In a complementary experiment, we employ `dxtb` (Friede et al., 2024) instead of neural approximators to obtain rewards and constraints, which offers exact gradients over atomic positions. For this experiment, we fine-tune FlowMol pre-trained on QM9 (Ramakrishnan et al., 2014). We again maximize the dipole moment while constraining the total energy to remain below $-18$ Ha, a value that differs from the constraint in the main paper due to the different atomic number distribution. As shown in Table 1, the pre-trained model $\pi^{pre}$ violates such a constraint with 65 % of samples. In contrast, the model fine-tuned via CFO can successfully achieve zero constraint violation (30 Monte Carlo samples, all below the threshold) while increasing the average norm of the dipole moment from $3.43 \pm 3.45$ to $8.66 \pm 4.50$, as shown in Fig. 8a. As a baseline comparison, we compare to just using Adjoint Matching (Domingo-Enrich et al., 2025), which increases the dipole to 9.04D but also the energy to $-15.5$Ha.

**Results using `posebuster` validity score function.** To further highlight CFO's flexibility, we replace the energy constraint with a molecular-validity criterion based on `posebuster` (Buttenschoen et al., 2024), while keeping the dipole moment as reward. We train a GNN on a custom validation score that equals zero when a molecule is connected and passes the basic `posebuster` checks, and 1 otherwise,

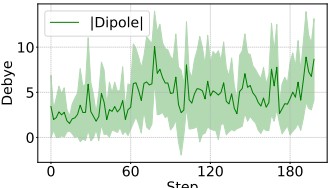

(a) Dipole Moment (in D)

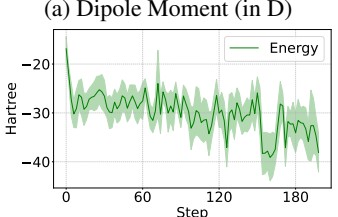

(b) Energy (in Ha)

Figure 8: Energy-constrained dipole moment maximization on QM9 (Ramakrishnan et al., 2014) and using `dxtb` (Friede et al., 2024) as reward and constraint functions, with exact gradients of the simulation.

Table 1: Numeric results for CFO on QM9 using `dxtb` for dipole and energy.

| Property | Stage | Value |
|---|---|---|
| Dipole moment | Pre | $3.43 \pm 3.45D$ |
| | CFO | $8.66 \pm 4.50D$ |
| Energy | Pre | $-16.72 \pm 2.48$ Ha |
| | CFO | $-39.40 \pm 4.01$ Ha |
| Violations | Pre | 65 % |
| | CFO | 0 % |

running CFO with $K = 2$, $N = 50$, and $B = 0.3$. The pre-trained model attains a dipole moment of 6.92 D but has a 53% constraint-violation rate. In contrast, CFO increases the reward to 9.60 D while reducing the predicted violation rate to 39%. In contrast to the energy constraints presented in the main text, the predicted violation rate also differs from the ground truth violation rate, which might be circumvented by an online learning of the constraint function.

**Additional Discussion on Validity of Molecules.** For the molecular design experiments on drug-like molecules presented in the main text, we further apply an RDKit validation step, including stereochemistry reassignment, hydrogen count correction, and full sanitization (valences, kekulization, bond orders). Approximately 7% of final molecules pass, which can be attributed to several reasons: Already in the base FlowMol model, only 34% of molecules fulfill the RDKit validation step, highlighting the need for more diverse pre-training datasets and further base model improvements. Furthermore, the FlowMol-generated geometries used during optimization are not geometrically relaxed, which can lead to invalid bond lengths or angles (see examples in Figure 9). This motivates the development of fully differentiable geometry relaxation methods for molecular design or the extension of CFO to different solvers.

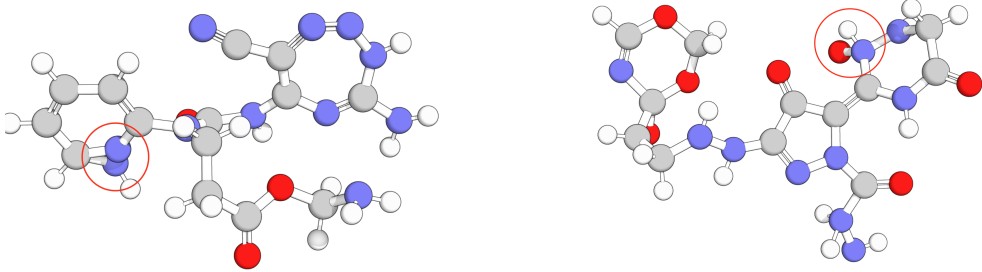

Figure 9: Generated drug-like molecules failing the validity test and showing unreasonable bond lengths and angles, highlighted with red circles.

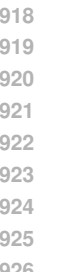
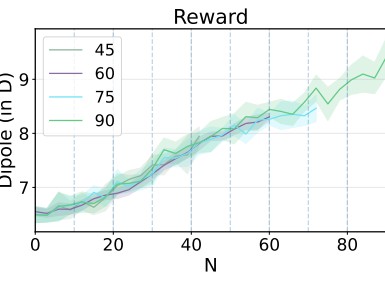
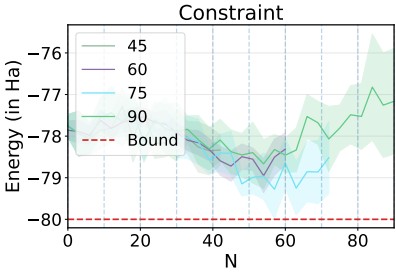

(a) Dipole (in D) for different $N$.

(b) Energy (in Ha) for different $N$.

Figure 10: Unconstrained Dipole maximization of AM (Domingo-Enrich et al., 2025), i.e., $\mu = 0$ in Eq. 7, for different $N$.

## D PARAMETER DETAILS AND ABLATION STUDIES FOR CONSTRAINED FLOW OPTIMIZATION AND ADJOINT MATCHING

Discussion of the most important Hyperparameter of CFO and FINETUNINGSOLVER:

- **Initial penalty** $\rho_{\text{init}}$**.** Larger $\rho_{\text{init}}$ penalizes constraint violations more strongly, thus effectively reducing early exploration inside the base distribution. Smaller $\rho_{\text{init}}$ does the opposite.

- **Penalty growth rate** $\eta \geq 1$**.** Controls the penalty growth across updates. Larger $\eta$ accelerates enforcement and thus can reduce exploration of high-reward regions. Smaller $\eta$ tightens feasibility more gradually, allowing for early reward progress, but potentially slower constraint satisfaction.

- **Contraction rate** $\tau \in (0, 1)$**.** Determines when the penalty parameter $\rho$ is updated. Smaller $\tau$ triggers more frequent updates, values near one update conservatively.

- **Multiplier lower bound** $\lambda_{\min} < 0$**.** Safeguards the Lagrange multiplier via clipping. Smaller $\lambda_{\min}$ permits larger corrective signals of the offset, see Sec. 4. If set to a large negative value, its influence on the final output is typically small, since $\lambda_{\min}$ is not achieved.

- **FINETUNINGSOLVER regularization** $\alpha$**.** Trade-off between staying close to the base distribution and reallocating mass. Larger $\alpha$ enforces stronger KL-regularization of the policy. A smaller $\alpha$ allows greater deviation from the base policy.

- **Sampling for constraint estimation (sample count/batch size).** Larger samples reduce estimator variance, stabilizing updates and improving feasibility. If the sample size is too small, this yields volatile or biased estimates that can steer CFO to off-target solutions.

Table 2: Hyperparameters for CFO and Adjoint Matching

|  | **SG**(2e-2g) | **MoG**(2a-2c) | **MD-QM9**(8a-8b) | **MD-GEOM**(3a-3b) |
|---|---|---|---|---|
| **CFO** | | | | |
| Lagrangian Updates $K$ | 20 | 20 | 20 | 6 |
| $\rho_{\text{init}}$ | 0.5 | 0.5 | 2 | 1.0 |
| $\eta$ | 1.25 | 1.25 | 1.1 | 1.25 |
| $\tau$ | 0.99 | 0.99 | 0.99 | 0.99 |
| $\lambda_{\min}$ | -50.0 | -50.0 | -50.0 | -50.0 |
| **Adjoint Matching** | | | | |
| $(1/\alpha)$ | 1e5 | 1e5 | 1e2 | 50 |
| Number of Iterations $N$ | 300 | 300 | 10 | 10 |
| Effective Batch Size | 256 | 256 | 40 | 20 |
| Clip Grad Norm | 0.7 | 0.7 | 0.5 | 0.4 |
| Learning Rate | 5e-6 | 5e-6 | 1e-4 | 5e-6 |
| Integration Steps | 40 | 40 | 50 | 40 |
| Total Steps | 6000 | 6000 | 200 | 60 |

**Ablation study for $\rho_{\mathbf{init}}, \eta, \tau,$ and $\lambda_{\mathbf{min}}$.** In the following, we provide an ablation study for the molecular design task (Figure 3a-3b) as well as the MoG task (Figure 2b).

Table 3: Ablation Study for MoG (2b) and Molecular Design Tasks (3a-3b)

| value | MoG Task | | Molecular Design Task | |
|---|---|---|---|---|
| | $\mathbb{E}[r(x)] \uparrow$ | $\mathbb{E}[c(x)] \downarrow$ | $\mathbb{E}[r(x)] \uparrow$ | $\mathbb{E}[c(x)] \downarrow$ |
| **PRE** | | | | |
| – | $-7.32 \pm 0.08$ | $0.58 \pm 0.02$ | $6.55 \pm 0.07$ | $-77.86 \pm 0.22$ |
| $\rho_{\text{init}}$ | | | | |
| 0.1 | $-4.49 \pm 0.06$ | $0.24 \pm 0.02$ | $8.43 \pm 0.12$ | $-82.21 \pm 0.33$ |
| 1.0 | $-4.88 \pm 0.07$ | $0.10 \pm 0.01$ | $8.36 \pm 0.11$ | $-81.99 \pm 0.45$ |
| 10.0 | $-5.62 \pm 0.09$ | $0.10 \pm 0.01$ | $8.22 \pm 0.08$ | $-81.91 \pm 0.37$ |
| $\eta$ | | | | |
| 1.0 | $-4.56 \pm 0.05$ | $0.17 \pm 0.01$ | $8.33 \pm 0.11$ | $-82.22 \pm 0.39$ |
| 1.25 | $-4.75 \pm 0.06$ | $0.12 \pm 0.01$ | $8.30 \pm 0.12$ | $-81.99 \pm 0.34$ |
| 2.0 | $-5.34 \pm 0.15$ | $0.10 \pm 0.01$ | $8.39 \pm 0.27$ | $-81.98 \pm 0.46$ |
| $\lambda_{\mathbf{min}}$ | | | | |
| 0.0 | $-4.84 \pm 0.04$ | $0.16 \pm 0.01$ | $8.39 \pm 0.10$ | $-81.85 \pm 0.31$ |
| -1.0 | $-4.40 \pm 0.76$ | $0.26 \pm 0.02$ | $8.30 \pm 0.12$ | $-81.80 \pm 0.42$ |
| -10.0 | $-4.75 \pm 0.06$ | $0.12 \pm 0.01$ | $8.26 \pm 0.11$ | $-82.13 \pm 0.39$ |
| -50.0 | $-4.75 \pm 0.06$ | $0.12 \pm 0.01$ | $8.35 \pm 0.13$ | $-82.16 \pm 0.48$ |
| $\tau$ | | | | |
| 0.5 | $-5.02 \pm 0.05$ | $0.10 \pm 0.01$ | $8.34 \pm 0.11$ | $-82.06 \pm 0.32$ |
| 0.75 | $-4.98 \pm 0.05$ | $0.10 \pm 0.01$ | $8.31 \pm 0.13$ | $-82.05 \pm 0.40$ |
| 0.9 | $-4.82 \pm 0.07$ | $0.11 \pm 0.01$ | $8.31 \pm 0.12$ | $-81.93 \pm 0.40$ |
| 0.99 | $-4.75 \pm 0.06$ | $0.12 \pm 0.01$ | $8.39 \pm 0.13$ | $-82.27 \pm 0.33$ |

Across tasks, CFO's sensitivity to hyperparameters varies: while the MoG task exhibits clear shifts in reward and constraint satisfaction across settings, the molecular design task remains highly robust, with only minor fluctuations. Larger initial $\rho_{\text{init}}$ and higher $\eta$ consistently tighten constraint satisfaction at the cost of modestly reduced reward, whereas $\lambda_{\text{min}}$ and $\tau$ have a lower effect. The effect is lower effect of $\lambda_{\text{min}}$ likely stems from $\lambda$ rarely reaching its lower bound, and the smoothing parameter barely impacts updates. A separate batch-size ablation on MoG shows that larger batches significantly improve constraint satisfaction and reward maximization.

Table 4: Ablation Study for the MoG task with different batch sizes

| value | $\mathbb{E}[r(x)] \uparrow$ | $\mathbb{E}[c(x)] \downarrow$ |
|---|---|---|
| Batch Size | | |
| 8 | $-5.16 \pm 0.11$ | $0.36 \pm 0.04$ |
| 32 | $-4.93 \pm 0.08$ | $0.27 \pm 0.05$ |
| 128 | $-4.74 \pm 0.06$ | $0.14 \pm 0.02$ |
| 512 | $-4.68 \pm 0.05$ | $0.11 \pm 0.01$ |

# E   PROOFS

Before we present a proof of the theorems in Section 5. We will transform the main problem in Eq. 5 to a simpler form. First, we recall that the policy $\pi$ is a vector field. It has been shown before that the ODE in Eq. 1 and a stochastic differential equation (SDE) of the form

$$dX_t = b(X_t, t)dt + \sigma(t)dB_t, \ X_0 \sim p_0, \tag{21}$$

with drift $b : \mathbb{R}^d \times [0, 1] \to \mathbb{R}^d$, diffusion coefficient $\sigma : [0, 1] \to \mathbb{R}_{\geq 0}$ and Brownian motion $B_t$ induce the same marginals $\{p_t\}$. For an exact definition of $b$ and a proof of this statement, we refer to (Domingo-Enrich et al., 2025). Controlling this SDE can be done by adjusting the drift as follows (Tang, 2024; Domingo-Enrich et al., 2025):

$$dX_t = \left(b(X_t, t) + \sigma(t)u(X_t, t)\right)dt + \sigma(t)dB_t, \ X_0 \sim p_0,$$

where $u : \mathbb{R}^d \times [0, 1] \to \mathbb{R}^d$ is a control vector field, this means the pre-trained model is a controlled model with $u \equiv 0$. With these notational changes, we reformulate the optimization problem in Eq. 5 in terms of the controlled diffusion process $\mathbf{X}^u \sim p^u$:

$$\max_{u \in \mathcal{U}} \quad \mathbb{E}_{\mathbf{X}^u \sim p^u}\left[r(X_1)\right] - \alpha D_{KL}(p_1^u(\cdot) || p_1^{\text{pre}}(\cdot)) \tag{22}$$
$$\text{s.t.} \quad \mathbb{E}_{\mathbf{X}^u \sim p^u}\left[c(X_1)\right] \leq B$$

Eq. 22 may seem the same as Eq. 5, but it is in terms of a diffusion process. This way we can calculate the KL efficiently, see (Eq. 18, Domingo-Enrich et al., 2025), by using Girsanov's theorem, which gives the relationship between the control process $u$ and the KL-Divergence:

$$D_{\text{KL}}(p^u(\mathbf{X}|X_0) \ || \ p^{\text{pre}}(\mathbf{X}|X_0)) = \mathbb{E}_{\mathbf{X}^u \sim p^u}\left[\int_0^1 \frac{1}{2}\|u(X_t, t)\|^2 \, dB_t\right]$$

Meaning if both processes have the same initial value $X_0$, the KL divergence between the controlled and uncontrolled process is equal to the expected value of the squared norm of the control $u$ (Domingo-Enrich et al., 2025; Uehara et al., 2024b; Tang, 2024). This dependence on the initial value can be dropped when using a specific noise schedule (Domingo-Enrich et al., 2025). Recalling that marginals at time $t$ are $p_t(x)$, i.e. $X_t \sim p_t(x)$, then we can equivalently write the optimization problem as:

$$\max_{u \in \mathcal{U}} \quad \mathbb{E}_{\mathbf{X}^u \sim p^u}\left[r(X_1)\right] - \alpha\mathbb{E}\left[\int_0^1 \frac{1}{2}\|u(X_t^u, t)\|^2 dt\right]$$
$$\text{s.t.} \quad \mathbb{E}_{\mathbf{X}^u \sim p^u}\left[c(X_1)\right] \leq B$$

Where the expectation is taken over the controlled process $\mathbf{X}^u$. For numerical optimization, we now assume that the control $u$ is a parametric model, typically a neural network, with parameters $\theta$. The resulting optimization problem is then:

$$\max_{\theta \in \mathbb{R}^m} \quad F(\theta) := F_r(\theta) - \alpha F_{KL}(\theta)$$
$$= \mathbb{E}_{x \sim p_1^{u_\theta}}\left[r(x)\right] - \alpha\mathbb{E}\left[\int_0^1 \frac{1}{2}\|u_\theta(X_t, t)\|^2 dt\right] \tag{23}$$
$$\text{s.t.} \quad G(\theta) := \mathbb{E}_{x \sim p_1^{u_\theta}}\left[c(x)\right] - B \leq 0$$

For some function $F : \mathbb{R}^m \to \mathbb{R}$ and function $G : \mathbb{R}^m \to \mathbb{R}$. This is finite-dimensional optimization over $\theta$.

Next, we present a proof that Alg. 1 can find a parameterized policy $\pi_\theta$, with $\theta \in \mathbb{R}^m$ that minimizes the infeasibility while maximizing the reward. The proof is adapted from "Practical Augmented Lagrangian Methods for Constrained Optimization" (Birgin & Martínez, 2014, Chapter 5).

The augmented Lagrangian objective in Eq. 8 becomes:

$$L_\rho(\theta, \lambda) = F(\theta) - \frac{\rho}{2}\left[\max\left(0, G(\theta) - \frac{\lambda}{\rho}\right)\right]^2 \tag{24}$$

where $\lambda \in \mathbb{R}_{\leq 0}$ is the Lagrange multiplier, $\rho > 0$ is a penalty parameter.

With this notation, the assumption on the solver becomes:

**Assumption E.1** (Solver). For all $k \in \mathbb{N}$, we obtain $u$ such that:

$$L_{\rho_k}(\theta_k, \lambda_k) \geq L_{\rho_k}(\theta, \lambda_k) - \varepsilon_k \quad \forall \, \theta \in \mathbb{R}^m \tag{25}$$

where the sequence $\{\varepsilon_k\} \subseteq \mathbb{R}_+$ is bounded.

This corresponds to Assumption 5.1 from (Birgin & Martínez, 2014). Assumption E.1 states that the solver can find an approximate maximizer of the subproblem.

Next we state and prove the main result for the algorithm. Namely, in the limit, we obtain a minimizer of the infeasibility measure.

**Theorem E.2** (Feasibility of **C**onstrained **F**low **O**ptimization). *Let $\{\theta_k\}$ be a sequence generated by Alg. 1 under the solver Assumption E.1. Let $\bar{\theta}$ be a limit of the sequence $\{\theta_k\}$. Then, we have:*

$$\langle G(\bar{\theta}) \rangle_+ \leq \langle G(\theta) \rangle_+ \quad \forall \theta \in \mathbb{R}^m, \tag{26}$$

*where $G(\theta) := \mathbb{E}_{x \sim p_1^{u_\theta}}[c(x)] - B \leq 0$ and $\langle \cdot \rangle_+ := \max\{0, \cdot\}$.*

*Proof.* By definition $\mathbb{R}^m$ is closed and $\theta_k \in \mathbb{R}^m$ thus $\bar{\theta} \in \mathbb{R}^m$. We consider two cases: $\{\rho_k\}$ bounded and $\rho_k \to \infty$. First we assume $\{\rho_k\}$ is bounded, there exists $k_0$ such that $\rho_k = \rho_{k_0}$ for all $k \geq k_0$. Therefore, for all $k \geq k_0$, the upper bracket of Eq. 13 holds. This implies that $|V_k| \to 0$, so $\langle G(\theta_k) \rangle_+ \to 0$. Thus, the limit point is feasible.

Now, assume that $\rho_k \to \infty$. Let $K \subseteq \mathbb{N}$ be such that:

$$\theta_k \to \bar{\theta} \text{ for } k \in K \text{ and } k \to \infty$$

Assume by contradiction that there exists $\theta \in \mathbb{R}^d$ such that

$$\langle G(\bar{\theta}) \rangle_+^2 > \langle G(\theta) \rangle_+^2$$

By the continuity of $G$, the boundedness of $\{\lambda_k\}$, and the fact that $\rho_k \to \infty$, there exists $c > 0$ and $k_0 \in \mathbb{N}$ such that for all $k \in K, k \geq k_0$:

$$\left\langle G(\theta_k) - \frac{\lambda_k}{\rho_k} \right\rangle_+^2 > \left\langle G(\theta) - \frac{\lambda_k}{\rho_k} \right\rangle_+^2 + c$$

Therefore, for all $k \in K, k \geq k_0$:

$$F(\theta_k) - \frac{\rho_k}{2} \left[ \left\langle G(\theta_k) - \frac{\lambda_k}{\rho_k} \right\rangle_+^2 \right] < F(\theta) - \frac{\rho_k}{2} \left[ \left\langle G(\theta) - \frac{\lambda_k}{\rho_k} \right\rangle_+^2 \right] - \frac{\rho_k c}{2} + F(\theta_k) - F(\theta)$$

Since $\lim_{k \in K} \theta_k = \bar{\theta}$, the continuity of $F$, and the boundedness of $\{\varepsilon_k\}$, there exists $k_1 \geq k_0$ such that, for $k \in K \; k \geq k_1$:

$$\frac{\rho_k c}{2} - F(\theta_k) + F(\theta) > \varepsilon_k$$

Therefore,

$$F(\theta_k) - \frac{\rho_k}{2} \left[ \left\langle G(\theta_k) - \frac{\lambda_k}{\rho_k} \right\rangle_+^2 \right] < F(\theta) - \frac{\rho_k}{2} \left[ \left\langle G(\theta) - \frac{\lambda_k}{\rho_k} \right\rangle_+^2 \right] - \varepsilon_k$$

for $k \in K, k \geq k_1$. This contradicts Assumption E.1. $\qquad \square$

Theorem E.2 and its proof correspond to (Birgin & Martínez, 2014, Sec. 5.1). Theorem E.2 establishes that Alg. 1, under the iterates given in Assumption E.1, identifies minimizers of the infeasibility, i.e.,

$$\langle G(\theta) \rangle_+ := \left\langle \mathbb{E}_{x \sim p_1^{u_\theta}}[c(x)] - B \leq 0 \right\rangle_+.$$

Consequently, if the original optimization problem is feasible, then every limit point of the sequence produced by the algorithm is also feasible.

Next, we will see that, assuming that $\varepsilon_k$ tends to zero, it is possible to prove that, in the feasible case, the algorithm asymptotically finds a global maximizer of the problem in Eq. 5.

**Theorem E.3** (**O**ptimality of **C**onstrained **F**low **O**ptimization). *Let $\{\theta_k\} \subset \mathbb{R}^d$ be a sequence generated by Alg. 1 under Assumption E.1 and $\lim_{k \to \infty} \varepsilon_k = 0$. Let $\bar{\theta} \in \mathbb{R}^m$ be a limit of the sequence $\{\theta_k\}$. Suppose that $\langle G(\theta) \rangle_+ = 0$, then $\bar{\theta}$ is a global maximizer of Eq. 5.*

*Proof.* Let $K \subseteq \mathbb{N}$ be such that.

$$\theta_k \to \bar{\theta} \ \text{ for } \ k \in K \ \text{ and } \ k \to \infty$$

By assumption, the problem is feasible, thus, by Theorem E.2, we have that $\bar{\theta}$ is feasible. Let $\theta \in \mathbb{R}^m$ be such that $G(\theta) \leq 0$. By the definition of the algorithm, we have that

$$F(\theta_k) - \frac{\rho_k}{2}\left[\left\langle G(\theta_k) - \frac{\lambda_k}{\rho_k} \right\rangle_+^2\right] \geq F(\theta) - \frac{\rho_k}{2}\left[\left\langle G(\theta) - \frac{\lambda_k}{\rho_k} \right\rangle_+^2\right] - \varepsilon_k \tag{27}$$

for all $k \in \mathbb{N}$, as well as by assumption $G(\theta) \leq 0$, we have that

$$\left\langle G(\theta) - \frac{\lambda_k}{\rho_k} \right\rangle_+^2 \leq \left(\frac{\lambda_k}{\rho_k}\right)^2. \tag{28}$$

We again consider the two cases: $\rho_k \to \infty$ and $\{\rho_k\}$ bounded.

In the first case, we assume $\rho_k \to \infty$. By Eq. 27 and Eq. 28, we have

$$F(\theta_k) \geq F(\theta_k) - \frac{\rho_k}{2}\left[\left\langle G(\theta_k) - \frac{\lambda_k}{\rho_k} \right\rangle_+^2\right] \geq F(\theta) - \frac{(\lambda_k)^2}{2\rho_k} - \varepsilon_k.$$

Taking limits for $k \in K$, and using that $\theta_k \to \bar{\theta}$, we have that $\lim_{k \in K} (\lambda_k)^2/\rho_k = 0$ and $\lim_{k \in K} \varepsilon_k = 0$, by the continuity of $F$ and the convergence of $\theta_k$, we get

$$F(\bar{\theta}) \geq F(\theta).$$

Since $\theta$ is an arbitrary feasible element of $\mathbb{R}^m$, $\bar{\theta}$ is a global optimizer.

For the second case, we assume $\{\rho_k\}$ is bounded, there exists $k_0 \in \mathbb{N}$ such that $\rho_k = \rho_{k_0}$ for all $k \geq k_0$. Therefore, by Assumption E.1, Eq. 27 holds for all $k \geq k_0$, and Eq. 28 holds with $\rho = \rho_{k_0}$. Thus,

$$F(\theta_k) - \frac{\rho_{k_0}}{2}\left[\left\langle G(\theta_k) - \frac{\lambda_k}{\rho_{k_0}} \right\rangle_+^2\right] \geq F(\theta) - \frac{(\lambda_k)^2}{2\rho_{k_0}} - \varepsilon_k.$$

for all $k \geq k_0$. Let $K_1 \subseteq \mathbb{N}$ and $\lambda^* \in \mathbb{R}_{\leq 0}$ be such that: $\lim_{k \in K_1} \lambda_k = \lambda^*$. By the feasibility of $\bar{\theta}$, taking limits in the inequality above for $\bar{k} \in K_1$, we get

$$F(\bar{\theta}) - \frac{\rho_{k_0}}{2}\left[\left\langle G(\bar{\theta}) - \frac{\lambda^*}{\rho_{k_0}} \right\rangle_+^2\right] \geq F(\theta) - \frac{(\lambda^*)^2}{2\rho_{k_0}} - \varepsilon_k. \tag{29}$$

Now, if $G(\bar{\theta}) = 0$, since $\lambda^*/\rho_{k_0} \geq 0$, we have that

$$\left\langle G(\bar{\theta}) - \frac{\lambda^*}{\rho_{k_0}} \right\rangle_+^2 = \left(\frac{\lambda^*}{\rho_{k_0}}\right)^2$$

Therefore, by Eq. 29,

$$F(\bar{\theta}) - \frac{\rho_{k_0}}{2}\left[\left\langle G(\bar{\theta}) - \frac{\lambda^*}{\rho_{k_0}} \right\rangle_+^2\right] \geq F(\theta) - \frac{(\lambda^*)^2}{2\rho_{k_0}}. \tag{30}$$

But, by Eq. 11, $\lim_{k \to \infty} \min\{G(\theta_k), -\lambda^*/\rho_{k_0}\} = 0$. Therefore, if $G(\bar{\theta}) < 0$, we necessarily have that $\lambda^* = 0$. Therefore, Eq. 30 implies that $F(\bar{\theta}) \geq F(\theta)$. Since $\theta$ is an arbitrary feasible element of $\mathbb{R}^m$, $\bar{\theta}$ is a global optimizer. $\qquad\square$

We want to make two remarks about Theorem E.3: first, as mentioned in Sec. 5, having access to such a solver is difficult and, in practice, rarely the case. Secondly, we refer the reader to (Birgin & Martínez, 2014, Sec. 5.2) for a discussion about the sets $K$ and $K_1$, how they are connected to the convexity of $F$ and $G$, and the corresponding theorem and proof.

