# OpenReview forum: "Constrained Generative Optimization via Progressive Flow Adaptation"
_ICLR.cc/2026/Conference — Submitted to ICLR 2026_

### Official Review · Reviewer_RqDF · 2025-10-20

**Soundness:** 2
**Presentation:** 3
**Contribution:** 2
**Rating:** 2
**Confidence:** 4

**Summary:**

The paper proposes Constrained Flow Optimization (CFO) as a method for fine-tuning pretrained generative models subject to explicit expectation constraints. CFO formulates the problem as a constrained optimization and applies an augmented Lagrangian approach to solve the posed problem. CFO is evaluated on both synthetic and real molecule design tasks.

**Strengths:**

- The ablation of the FineTuningSolver algorithm in line 414 is helpful in being able to understand how the performance of CFO is affected by the FineTuningSolver.
- The manuscript is generally well-written and the proposed CFO algorithm is intuitive and easy to understand.

**Weaknesses:**

1. Assumption 5.1 is a relatively strong assumption that requires the FineTuningSolver to return near-globally-optimal policies, and Theorem 5.4 further assumes $\varepsilon_k\to 0$ as $k\to\infty$ (i.e., arbitrarily accurate inner solves). These are unrealistic assumptions that are almost certainly violated in real-world settings with practical deep networks and standard optimizers - even if this is noted to be a previously held assumption in prior work (line 271). While I appreciate that this limitation is acknowledged in line 296, this limitation essentially invalidates the relevance of much of the theoretical contributions and convergence guarantees to the empirical setup. The manuscript should either provide empirical evidence that these assumptions are valid (e.g., in the synthetic tasks assuming the global optima are known) or consider proving a version of Theorem 5.4 without making use of especially strong assumptions.
2. There are a number important ablation studies that are missing. More specifically, it is unclear how sensitive the proposed CFO method is with repect to (1) the initial penalty parameter $\rho_{\text{init}}$; (2) the penalty growth rate parameter $\eta$; (3) the contraction parameter $\tau$; and (4) the minimum Lagrant multiplier $\lambda_{\text{min}}$. These ablations are particularly important since Algorithm 1 seems to be heuristic-based.
3. I think it is also important to independently (1) ablate Step 4 in Algorithm 1 and set the Lagrange multiplier to a constant value independent of $k$ (and showing how CFO performance changes as a function of this constant value); and (2) ablate Step 5 in Algorithm 1 and set the penalty $\rho$ to a constant value independent of $k$ (also modulate the exact value of this constant in the ablation). I note that the first ablation in this comment is essentially the baseline approach mentioned by the authors in line 54 to include constraint(s) as manually weighted objective function(s); the goal would be to empirically demonstrate that CFO is indeed an improvement over this baseline.
4. REINFORCE and rejection sampling also seem to be relevant baselines that should be included to evaluate empirically against.
5. In Figure 4(b), it looks like the AM baseline has not converged yet - it's unclear if the AM method will eventually satisfy the constraint if given enough steps.
6. The optimization is formulated as an expectation constraint in Equation 5. That is a much weaker requirement than per-sample feasibility or even a PAC guarantee, and the specific application for molecule design would seem like it's especially important to achieve per-sample guarantees (e.g., toxicity, chemical validity).
7. I'm not sure why the manuscript specifically focuses on only synthetic and molecular design as the application areas of CFO. In principle, CFO seems to be broadly applicable to any generative optimization problem - and indeed, this is how it sounds from the title of this work. Especially since there are a number of hyperparameters involved with CFO, it would be important to show that CFO performance is robust across different tasks with the same hyperparameter configurations.

**Questions:**

8. In line 219, why is the initial Lagrange multiplier value set to 0 and not $\lambda_{\text{min}}$?
9. What if there are multiple constaints in Equation (5)? Importantly, what happens if the constraint(s) do not admit a feasible set?
10. It seems like [this paper](https://proceedings.mlr.press/v267/yao25b.html) considers an almost identical problem formulation (albeit different empirical evaluation setup) as in Equation (5) and manages to arrive at an exact solution for the problem that does not rely on heuristics as Algorithm 1 does. The only different seems to be the choice of constraint function and the definition of the pre-trained policy. How does the proposed approach vary from/improve upon this prior work?

---

> ### Author Response · Authors · 2025-12-02
>
> We thank the Reviewer for the constructive feedback and the positive comments regarding the manuscript’s clarity and the intuition underlying CFO. We address the points below.
>
> $\textbf{Weakness:}$
>
> $\textbf{(1) Strong assumptions in Assumption 5.1 and Theorem 5.4.}$
>
> These idealized assumptions follow the standard oracle model in augmented-Lagrangian analyses and cleanly separate the outer updates from solver-specific behavior, consistent with prior work [1, 2]. Reviewer jsKV emphasized the importance of providing formal guarantees, which our analysis delivers. Empirically, CFO performs well even when these assumptions are violated, indicating that coarse improvement steps suffice in practice. Extending the theory to weaker approximations is possible but constitutes a separate theoretical effort.
>
> $\textbf{(2) Missing ablations on $\rho_\text{init}, \eta, \tau,$ and $\lambda_\text{min}$.}$
>
> We have added ablation studies for a broad range of values for both major experiments in Appendix D. We observe that CFO’s performance remains robust across a wide range of parameter settings in the molecular design experiment. For the illustrative examples, the parameters have the following influence: Small values of $\rho_\text{init}$ or $\eta$ slow constraint enforcement, whereas excessively large values make the constraint dominate too early and impede reward optimization. The overall conclusions of the paper are unchanged.
>
> $\textbf{(3) Ablating Steps 4 and 5 (constant $\rho$ and $\lambda$).}$
>
> We have included a constant $\lambda=0.0$ ablation in the general ablation (this does not violate the assumptions of the proof; one may set $\lambda_\text{min}=0.0$). We have also included an ablation with fixed $\rho$, where we set $\eta = 1.0$ (please note this violates the assumptions of the proof). Neither ablation produces qualitatively different behavior from the main results.
>
> $\textbf{(4) REINFORCE and rejection sampling baselines.}$
>
> Recent work on diffusion-model adaptation has shown that REINFORCE-style updates (e.g., DDPO [3]), which optimize the reward without KL regularization, are generally less stable and more prone to mode collapse than KL-regularized approaches. For this reason, KL-regularized methods such as AM [4] are more stable and empirically superior to purely reward-maximizing alternatives in diffusion-model fine-tuning.
>
> For rejection sampling, we add an experiment using the illustrative example in Appendix B. We fine-tune a policy purely on the reward signal using Adjoint Matching and then enforce feasibility only by rejecting samples that violate the constraint. On the example in Figure 2c, this reward-only policy attains a constraint satisfaction rate of $13.40$ \%, compared to $84.40$ \% for the policy fine-tuned with CFO, i.e., accounting for the constraint during fine-tuning. Closer analysis indicates that (1) CFO’s rare violations are concentrated near the boundary, while (2) rejection sampling breaks down when the reward peak and feasible set are misaligned
>
> $\textbf{(5) Convergence of the AM baseline in Figure 4(b).}$
>
> The earlier version of Figures 4(a) and (b) did not clearly convey the intended comparison. Therefore, we have updated them together with the caption and the text.
> The correct presentation is as follows (see reply to jsKV): there exists a theoretical optimal $N^* $ for which AM finds the maximum for any given (augmented) reward. We find this maximum to be $N^* = 60$; thus, we show a constant line for AM with the reward and constraint after $N^* = 60$ steps, because AM is constant w.r.t. $K$.
>
> For CFO, we find that an approximate oracle works well, and we choose $K=6$ with $N=10$, i.e., each set of parameters $\lambda$ and $\rho$ defines the reward for $10$ FineTuningSolver calls. This leads to two conclusions: (1) CFO can work with an approximate solver and still perform well, and (2) CFO has a similar computational burden as the FineTuningSolver (see last paragraph of Sec. 6).
>
> In Appendix D, we added a plot showing the evolution of AM for different $N$.
>
> $\textbf{(6) Expectation constraint vs per-sample / PAC guarantees.}$
> We agree that an expectation constraint is weaker than per-sample feasibility or PAC-style guarantees. In our work, we focus on the widely used expectation formulation, which is standard in constrained policy optimization and already represents a significant step over unconstrained fine-tuning. Nonetheless, CFO can be adapted to stronger forms of constraint, e.g., by applying a ReLU to enforce $\mathbb{E}[\mathrm{ReLU}(c(x)-B)] \le 0$, following [e.g., 5].

---

> ### Author Response · Authors · 2025-12-02
>
> $\textbf{(7) Application domains and robustness across tasks.}$
>
> We have updated the paper title to clarify that our primary focus is molecular design. This domain offers a high-dimensional setting with well-defined property constraints, making it well-suited for isolating the effect of constraint generative optimization. Although CFO is general and could be applied to other modalities, molecular design provides a compelling and representative testbed that captures core challenges. We agree that exploring additional domains is an interesting direction for future work; however, the current experiments already reflect the key difficulties of constrained generation and adequately demonstrate CFO’s capabilities.
>
> $\textbf{Questions:}$
>
> $\textbf{(1) Why is the initial Lagrange multiplier set to 0 and not $\lambda_\text{min}$?}$
>
> We intentionally initialize $\lambda_0 = 0$ to start with a very weak constraint penalty, allowing the algorithm to first adapt to the objective landscape and then gradually increase the penalty as violations persist. The parameter $\lambda_\text{min}$ in CFO is used as a lower bound to prevent $\lambda$ from going to $\infty$ during the updates, rather than as a meaningful prior on the “correct” penalty weight. Initializing with $\lambda_0 = \lambda_\text{min}$ would impose a stronger penalty from the beginning and could reduce early exploration. In fact, one may initialize as $\lambda_0 \in [\lambda_\text{min}, 0]$ without violating the assumptions of the proof.
>
> $\textbf{(2) Multiple constraints and infeasible constraint sets.}$
> An extension to multiple constraints is straightforward: one introduces a Lagrange multiplier $\lambda^j$ and penalty $\rho^j$ for each constraint $c^j$, leading to the multi-constraint augmented reward:
>
> $
> f_k(x) = r(x) + \sum_j \frac{\rho_k^j}{2} \left[ \max \left( 0, c^j (x) - B^j - \frac{\lambda_k^j}{\rho_k^j} \right) \right]^2
> $
>
> CFO then maintains a vector of multipliers and penalties and updates each coordinate analogously [1, 6].
> When no feasible solution exists, CFO converges to the least-infeasible solution (Theorem 5.1), minimizing a weighted combination of constraint violations.
>
> $\textbf{(3) Relation to DynAMO.}$
>
> We thank the Reviewer for highlighting this work. The proposed method DynAMO [7] optimizes latent design variables directly using an offline surrogate and a distribution-matching regularizer.
> While the problems share a similar structure, DynAMO targets latent-space optimization rather than fine-tuning a generative model to enforce constraints. CFO addresses a different setting: we update the diffusion model itself to maximize reward while enforcing constraints. Because the optimization variables and objective structure differ fundamentally, the two approaches are not comparable and solve distinct problems.
> DynAMO’s exact solution applies to its latent-space objective and does not immediately clear how to extend to the parameter-space optimization required for constrained generative-model fine-tuning.
>
>
> $\textbf{References:}$
>
> [1] E. G. Birgin and J. M. Martinez. Practical Augmented Lagrangian Methods for Constrained Optimization. Fundamentals of Algorithms. Society for Industrial and Applied Mathematics, May 2014. ISBN 978-1-61197-335-8.
>
> [2] Riccardo De Santi, Marin Vlastelica, Ya-Ping Hsieh, Zebang Shen, Niao He, and Andreas Krause. Flow density control: Generative optimization beyond entropy-regularized fine-tuning. In The Exploration in AI Today Workshop at ICML 2025
>
> [3] Kevin Black, Michael Janner, Yilun Du, Ilya Kostrikov, & Sergey Levine. (2024). Training Diffusion Models with Reinforcement Learning.
>
> [4] Carles Domingo-Enrich, Michal Drozdzal, Brian Karrer, and Ricky T. Q. Chen. Adjoint Matching: Fine-tuning Flow and Diffusion Generative Models with Memoryless Stochastic Optimal Control, January 2025.
>
> [5] Li, J., Fridovich-Keil, D., Sojoudi, S., & Tomlin, C. (2021). Augmented Lagrangian Method for Instantaneously Constrained Reinforcement Learning Problems. In 2021 60th IEEE Conference on Decision and Control (CDC) (pp. 2982–2989).
>
> [6] Luiz F. O. Chamon, Mohammad Reza Karimi, and Anna Korba. Constrained sampling with primal-dual langevin monte carlo, 2025.
>
> [7] Yao, M., Gee, J., & Bastani, O. (2025). Diversity By Design: Leveraging Distribution Matching for Offline Model-Based Optimization. In Proceedings of the 42nd International Conference on Machine Learning (pp. 71687–71738). PMLR.

---

### Official Review · Reviewer_6dLh · 2025-10-29

**Soundness:** 4
**Presentation:** 3
**Contribution:** 2
**Rating:** 2
**Confidence:** 4

**Summary:**

This work presents Constrained Flow Optimization (CFO), a framework for optimizing pre-trained generative flow models that simultaneously maximizes rewards while satisfying explicit constraints. Using an Augmented Lagrangian approach, CFO automatically handles the balance between reward optimization and constraint adherence without requiring manual parameter tuning. The authors establish theoretical convergence properties and validate the approach through synthetic experiments and molecular design applications, showing reliable constraint enforcement compared to baseline methods.

**Strengths:**

The authors clearly identify a gap in current generative fine-tuning approaches—namely, the lack of principled handling of hard constraints in optimization problems. Introducing the augmented Lagrangian method into this constrained optimization problem is well motivated and sound. And the paper is well organized, with detailed pseudocode, clear notations, and reproducibility notes. Appendices provide implementation details, dataset descriptions, and solver pseudocode.

**Weaknesses:**

My primary concern is that the technological contribution appears incremental. The proposed algorithm largely applies the classical augmented Lagrangian framework to fine-tune a flow-based model. The main difficulty lies in practical tuning rather than in introducing a fundamentally new theoretical concept. Although the authors provide a theoretical analysis, it does not substantially advance the state of the art from a methodological standpoint.

In addition, the selection of baselines is rather limited. The comparisons are primarily made against a pretrained model without fine-tuning and the AM method. This choice seems designed to highlight the method’s novelty rather than to demonstrate its practical utility. Including additional baselines—such as methods employing soft constraints, test-time optimization strategies, or other comparable approaches—would strengthen the empirical evaluation and better substantiate the practical value of the proposed technique.

**Questions:**

See weakness.

---

> ### Author Response · Authors · 2025-12-02
>
> We thank the Reviewer for the constructive feedback and for highlighting the clarity of our presentation and pseudocode.
>
> We first note that in generative optimization via RL/control, it is standard and scientifically valuable to adapt classical optimization tools to the stochastic, sampling-based structure of modern generative models. For example:
> - Entropy regularized MDPs and RL [e.g., 1, 2] $ \rightarrow $ Entropy regularized fine-tuning [3],
> - Mirror Descent [4] $ \to $ Flow density control [5],
> - Chance Constrained Programming [e.g., 6] $ \to $ A Gradient Guided Diffusion Framework for Chance Constrained Programming [7]).
>
> Our work follows this paradigm and extends it by addressing the problem of reward-guided fine-tuning of flow and diffusion models under explicit hard constraints (constrained generative optimization), for which no prior work exists to the best of our knowledge.
>
> While augmented Lagrangian (AL) methods are classical, our work is the first, to the best of our knowledge, to demonstrate their applicability for fine-tuning flow/diffusion models under constraints. The adaptation is technically meaningful, as it requires handling the stochastic and sampling-driven dynamics of the underlying generative models.
> Other Reviewers independently recognized this as a novel contribution (jsKV, strength 1) and emphasized the intuitiveness of our approach (RqDF, strength 2).
>
> Importantly, methodological difficulty should not be a criterion for scientific merit. Simplicity that enables stable and effective optimization, as supported by our experiments (Section 6), is often a key strength.
>
> Regarding the request for additional baselines, we note that our method is designed specifically for fine-tuning of flow and diffusion models under hard constraints. Approaches based on soft constraints optimize a different objective, and test-time optimization methods usually operate in a different regime. The Reviewer does not provide references for the suggested alternatives, making it difficult to assess which specific methods they had in mind or how they would meaningfully compare in our setting. Another Reviewer also pointed out that identifying directly comparable baselines for this new problem formulation is nontrivial (e.g., jsKV).
>
> We share the Reviewer’s interest in enriched comparisons and have therefore incorporated a fixed-weight Lagrangian baseline, following the suggestions from other Reviewers (e.g., jsKV). The baseline and its quantitative evaluation are shown in Section 6 (especially Figure 5), which demonstrates that our observation from Section 3 holds: one needs very careful tuning of the fixed parameter $ \mu $, whereas our adaptive method performs well across parameters (see the new ablation in Appendix D).
>
> $ \textbf{References} $
>
> [1] Brian D Ziebart, Andrew L Maas, J Andrew Bagnell, Anind K Dey, et al. Maximum entropy inverse reinforcement learning. In Aaai, volume 8, pages 1433–1438. Chicago, IL, USA, 2008.
>
> [2] Gergely Neu, Anders Jonsson, & Vicenç Gómez. (2017). A unified view of entropy-regularized Markov decision processes.
>
> [3] Masatoshi Uehara, Yulai Zhao, Kevin Black, Ehsan Hajiramezanali, Gabriele Scalia, Nathaniel Lee Diamant, Alex M. Tseng, Tommaso Biancalani, and Sergey Levine. Fine-Tuning of Continuous-Time Diffusion Models as Entropy-Regularized Control, February 2024.
>
> [4] Arkadi Nemirovsky and David Yudin. Problem Complexity and Method Efficiency in Optimization. John Wiley & Sons, 1983
>
> [5] Riccardo De Santi, Marin Vlastelica, Ya-Ping Hsieh, Zebang Shen, Niao He, and Andreas Krause. Flow density control: Generative optimization beyond entropy-regularized fine-tuning. In The Exploration in AI Today Workshop at ICML 2025
>
> [6] Aharon Ben-Tal and Arkadi Nemirovski. Robust solutions of linear programming problems contaminated with uncertain data. Mathematical Programming, 88:411–424, 2000.
>
> [7] Boyang Zhang, Zhiguo Wang, & Ya-Feng Liu. (2025). A Gradient Guided Diffusion Framework for Chance Constrained Programming.

---

### Official Review · Reviewer_jsKV · 2025-10-30

**Soundness:** 3
**Presentation:** 3
**Contribution:** 3
**Rating:** 6
**Confidence:** 3

**Summary:**

This paper addresses the critical problem of Constrained Generative Optimization (CGO), where a generative model (specifically, a flow model) must be adapted to maximize an objective function while satisfying another hard constraint. Common fine-tuning based methods usually fail to obey hard constraints. The proposed CFO transforms the constrained problem into a sequence of unconstrained, KL-regularized fine-tuning subproblems. By progressively and adaptively tuning the Lagrange multiplier and penalty parameter, CFO automatically and provably balances the trade-off between reward maximization and constraint satisfaction. Experimental validations are provided.

**Strengths:**

1. The introduction of Constrained Flow Optimization (CFO), which leverages the powerful and robust Augmented Lagrangian scheme to solve the CGO problem, is novel and sound.
2. The paper provides formal convergence guarantees, which is a major advantage over heuristic methods.
3. The low-dimensional (2D) experiment part has good organization and is also illustrative (i.e., Fig 2 and 3). The authors have made the important message easy to follow, thanks to its clear presentation.

**Weaknesses:**

1. While Figure 4 presents results for the crucial molecular design task, the comparison is not abundant. A full evaluation should include results comparing the CFO against more SOTA baselines as well as common heuristic baselines. While I understand widely acknowledged baselines in the area of the CGO problem might be limited, one must-have heuristic baseline is the standard fixed-weight Lagrangian, which can also serve as an ablation study to verify that the progressive tuning effectively improves over a fixed-weight counterpart.
2. While formal theories about the adaptive Lagrangian optimization are provided, I found the writing has much space for improvement. For example, the entire Section 5 proceeds too quickly without sufficient clarification. Eq. 14, 15, and Theorem 5.4 are especially hard to parse with non-trivial notations. It is also not clear what the implications are for most of the content in the section, except the final outcome: CFO enjoys global optimality under certain conditions.

**Questions:**

1. How was the number of MC iterations for the constraint and reward estimations determined in the high-dimensional setting? Is it possible to provide a theoretical and/or logical analysis showing how increasing the number of MC iterations might reduce the small violations observed in the results?
2. Considering deviations, it seems Figure 4c does not really prove CFO's advantage over AM? Note AM does not need progressive optimizaiton so it consumes significantly less computation overheads. About this part it is also important to add computation analysis between CFO, PRE and AM because non-significant performance improvement becomes less promising at the price of costs.
3. It is highly recommended to improve theory section's writing to improve readability. Even not going too deep into details, it's important to have a quick take-away overview of what the notations are meaning and how they are connected to each other to finally prove global optimality,

---

> ### Author Response · Authors · 2025-12-02
>
> We thank the Reviewer for the constructive feedback and for highlighting the novelty and soundness of our proposed algorithm. We address each of the Reviewer’s comments individually below.
>
> $\textbf{Weakness:}$
>
> $\textbf{Lack of heuristic/fixed Lagrangian weight baseline.}$
>
> As the Reviewer correctly noted, there is no well-established baseline for constrained generative optimization in this setting, to the best of our knowledge. Following the Reviewer’s suggestion, we added a fixed-weight Lagrangian baseline to the revised manuscript. The new results appear in Section 6 (Figures 5a and 5b). They confirm the observation discussed in Section 3: performance is highly sensitive to the choice of the fixed penalty parameter $\mu$, whereas the adaptive update used in CFO is robust across a broad range of values (see also the new ablation in Appendix D). We appreciate the Reviewer’s guidance here and agree that including this heuristic baseline strengthens the empirical evaluation.
>
> $\textbf{Improved writing of Section 5.}$
>
> We substantially revised Section 5 to improve both readability and conceptual structure. The section now clearly explains the implications of the theorems and corollary, with the new clarifying text highlighted in blue.
>
> $\textbf{Questions:}$
>
> $\textbf{Meaning of “MC numbers”.}$
>
> The Reviewer’s question suggests a misunderstanding regarding the phrase “MC iterations.”, since there is no mention of “MC iterations” in our original manuscript.
>
> We interpret the Reviewers' meaning of “MC iterations” to refer to the number of Monte Carlo samples used to approximate the expected constraint in Eq. 11 (Step 3 of CFO).
>
> All samples are drawn i.i.d. from the same policy $\pi_k$, enabling the use of standard high-dimensional concentration inequalities (e.g., Hoeffding’s inequality), allowing us to quantify the accuracy of the empirical constraint estimate as a function of the number of samples.
> In practice, we select the number of samples to be as large as feasible on a single GPU; in our experiments, we use #Samples = 100, and we observe that performance is stable across deviations from this choice.
>
> Should the Reviewer be referring to a different notion of ‘MC iterations,’ we are glad to elaborate further.
>
>
> $\textbf{Clarification regarding AM’s convergence behavior and computational cost (Figure 4).}$
> To address the Reviewer’s concern, we made two clarifications:
>
> $\textit{1. Feasibility:}$
> CFO produces a constraint-satisfying policy, unlike AM, which demonstrates CFO’s advantage in ensuring feasibility.
>
> $\textit{2. Matched computational budgets:}$ We now explicitly clarify how we match computational budgets.
>
> - $\textbf{PRE:}$ the pretrained model $\pi_0$ with no additional cost.
>
> - $\textbf{AM:}$ There is a theoretical optimal number $N^* $ such that AM finds the optimum for every given (augmented) reward. We find $N^*= 60$ works well; see the newly added plots in
>
> - $\textbf{CFO:}$ Within CFO, we use AM as the FineTuningSolver. We run CFO for $K (=6)$ iterations. Each iteration performs one AM call with $N=10$, i.e., we use an AM as an approximate Solver.
>
> Under this setup, CFO and AM have equivalent computational cost, as both perform $60$ gradient steps. Concretely, CFO has a total runtime of $37.18$ min and compares well to the runtime of AM with $35.35$ min. The 5 \% overhead stems from the additional sampling and constraint evaluation required in Step 3 of CFO. We have updated Figure 4 together with the caption to make the outperformance clearer and avoid potential confusion.
>
> These updates make both AM’s convergence properties and the fairness of the computational comparison with CFO explicit.
>
> $\textbf{Comment on writing.}$
>
> See Weakness 2.

---

### Official Review · Reviewer_5dqt · 2025-11-01

**Soundness:** 3
**Presentation:** 3
**Contribution:** 3
**Rating:** 6
**Confidence:** 2

**Summary:**

In this work, the authors introduce Constrained Flow Optimization (CFO) which is a principled framework for fine-tuning generative flow or diffusion models under explicit constraints. The central problem addressed is how to adapt pre-trained generative foundation models (e.g., FlowMol, diffusion backbones) to maximize task-specific rewards (e.g., molecular property improvement) while ensuring constraint satisfaction (e.g., synthesizability or energy bounds).

The authors first formalize this setting as Constrained Generative Optimization (CGO), bridging ideas from optimization and generative modeling. They then derive CFO as a dual optimization algorithm based on the augmented Lagrangian method, which transforms the constrained objective into a series of standard KL-regularized fine-tuning problems. This approach allows progressive adaptation of generative models without manual tuning of reward–constraint trade-offs and provides theoretical guarantees for convergence and constraint satisfaction.

Empirically, CFO is evaluated on synthetic 2D settings and high-dimensional molecular design tasks, demonstrating that it can increase rewards (e.g., dipole moment) while enforcing stability constraints (e.g., xTB energy bounds). The authors further show CFO’s compatibility with differentiable and non-differentiable constraint functions, highlighting its generality for scientific discovery domains such as drug and material design.

**Strengths:**

The authors effectively demonstrate that Constrained Flow Optimization (CFO) scales to realistic, high-dimensional molecular design problems. Applying the method to a nontrivial chemical property optimization task of maximizing dipole moment under energetic stability constraints provides a credible and practically relevant benchmark.

**Weaknesses:**

With regards to the molecular design task, while CFO demonstrates effective scalar property optimization, the generated molecules show reduced chemical validity and unclear structural diversity, both of which are critical in molecular design and drug discovery. This raises concerns that the observed gains may stem from model bias exploitation or surrogate inaccuracies, rather than genuinely improving molecular quality. Although the authors briefly acknowledge this limitation, a more systematic analysis such as evaluating scaffold diversity, Lipinski or QED distributions, and functional group statistics would strengthen claims of chemical realism and practical relevance.

Furthermore, the paper would benefit from comparisons with diversity-preserving fine-tuning methods, such as Relative Trajectory Balance (RTB) (https://arxiv.org/abs/2503.06337, https://arxiv.org/pdf/2503.06337). RTB explicitly mitigates mode collapse and maintains distributional breadth during optimization, which is especially relevant for molecular generation tasks where balancing property improvement and diversity is crucial. Including such baselines would clarify whether CFO’s improvements come at the cost of diversity and robustness.

**Questions:**

Please see weaknesses

---

> ### Author Response · Authors · 2025-12-02
>
> We thank the Reviewer for the constructive feedback and for highlighting the relevance of our molecular design setting. We respond to each of the reviewer’s points in detail in the following:
>
> $\textbf{Molecular validity and diversity.}$
>
> We have expanded the evaluation to include validity rates, QED, Lipinski scores, and Murcko scaffolds, now reported in Section 6 (Table 5c). Compared to AM, CFO generates more valid molecules while maintaining similar QED and Lipinski values. We attribute the observed shifts in logP and related descriptors to the optimization objective (maximizing dipole) rather than to limitations of CFO. Together, these metrics illuminate how the optimization target influences molecular properties and offer a more complete evaluation of the resulting policies.
>
> $\textbf{RTB and diversity-preserving baselines.}$
>
> We appreciate the reviewer’s suggestion to consider Relative Trajectory Balance (RTB). Although originally developed for GFlowNets, subsequent work has shown that RTB can be applied to flow and diffusion models [1], and further that RTB is equivalent to Trust-PCL [2], an entropy-regularized control/RL method. Since Adjoint Matching optimizes the same underlying entropy-regularized control objective, both RTB and AM yield the same closed-form solution when used as a fine-tuning oracle. Thus, RTB could in principle serve as an alternative FineTuningSolver within CFO. While this is not essential to CFO’s formulation, it may offer an interesting avenue for future exploration. In this work, we compare CFO directly to AM (Section 6, Figure 4) and report the extended diversity and property metrics described above.
>
> $\textbf{References}$
>
> [1] Siddarth Venkatraman, Moksh Jain, Luca Scimeca, Minsu Kim, Marcin Sendera, Mohsin Hasan, Luke Rowe, Sarthak Mittal, Pablo Lemos, Emmanuel Bengio, Alexandre Adam, Jarrid Rector-Brooks, Yoshua Bengio, Glen Berseth, & Nikolay Malkin. (2025). Amortizing intractable inference in diffusion models for vision, language, and control.
>
> [2] Tristan Deleu, Padideh Nouri, Yoshua Bengio, & Doina Precup. (2025). Relative Trajectory Balance is equivalent to Trust-PCL.
>
> [3] Carles Domingo-Enrich, Michal Drozdzal, Brian Karrer, and Ricky T. Q. Chen. Adjoint Matching: Fine-tuning Flow and Diffusion Generative Models with Memoryless Stochastic Optimal Control, January 2025.

---

### Author Response · Authors · 2025-12-02

We thank all Reviewers for their thoughtful and constructive feedback. Several strengths of the submission were highlighted: the clear motivation and organization of the paper, including the identification of a gap in principled handling of hard constraints and the clarity of the presentation (6dLh, RqDF); the soundness of the overall optimization framework based on the augmented Lagrangian approach (6dLh, jsKV); the formal convergence guarantees and the illustrative low-dimensional experiments (jsKV); and the empirical demonstration that the CFO algorithm scales to realistic, high-dimensional molecular design tasks (5dqt). We appreciate the Reviewers’ careful evaluation and thorough examination of our manuscript.

The main concern raised by the Reviewers was the limited empirical evaluation of our method. In response, we have substantially revised the manuscript and used the extra page allowed by the ICLR guidelines to add one full page of new quantitative results directly addressing this concern and related limitations.

Our revisions introduce the following key additions and improvements:

$\textbf{1. New baseline with quantitative evaluation.}$

As suggested by Reviewers jsKV and RqD, we introduced an additional baseline using a fixed-$\mu$ Lagrangian parameter and report results in Section 6 (Figs. 5a-5b). Consistent with our observation in Section 3, performance is highly sensitive to the choice of $\mu$.

$\textbf{2. Expanded ablation studies across CFO parameters.}$

We perform ablations over $\mu$ (baseline), $\rho_\text{init}$, $\eta$, $\tau$, $\lambda_\text{min}$, and the number of Monte Carlo samples (Appendix D). We find that CFO achieves strong performances with diverse parameterizations in the molecular design task, and illustrate how extreme parameter choices (e.g., too small $\rho_\text{init}$ or $\eta$) mainly affect the speed of constraint.

$\textbf{3. Additional evaluation metrics for molecular design.}$

We added validity, QED, Lipinski, and Murcko scaffolds distributions metrics, as requested by Reviewer 5dqt, as well as the logP values (Section 6, Fig. 5c), as requested by Reviewer 5dqt. CFO generates twice as many valid molecules as AM, despite the expected drop in validity when aggressively optimizing dipole moment. QED/Lipinski remain stable, and $\log$P decreases, reflecting a natural reward–constraint tradeoff rather than a flaw of CFO.

$\textbf{4. Improved Presentation.}$

We update Figure 4 to avoid misleading interpretations and now explicitly compare computational budgets: CFO ($K=6$, $N=10$) and AM ($N^*=60$) use the same number of gradient steps, and we report the corresponding runtimes (37.18 min vs. 35.35 min; ~5% overhead). We substantially rewrote Section 5 to improve clarity in line with Reviewer jsKV’s feedback. In addition, we include the requested rejection-sampling baseline (RqDF) for comparison. On the same example, rejection sampling satisfies only 13.40% of constraints, while CFO achieves 84.40%, highlighting the clear advantage of constraint-aware fine-tuning.

$\textbf{5. Revised the title of the paper.}$

We revised the title to highlight our focus on molecular design:

- $\textbf{From: Constrained Generative Optimization via Progressive Flow Adaptation}$
- $\textbf{To: Constrained Flow Optimization via Sequential Fine-Tuning for Molecular Design}$

Molecular design offers a high-dimensional setting with clear property constraints, making it an ideal and highly practical relevant testbed for evaluating constrained optimization methods. Although CFO is broadly applicable, our experiments and contributions center on this domain. For this reason, we updated the title to more accurately reflect the scope and emphasis of the work.

We believe that with these new experimental results, further validation, and increased quality, we tackled the main concerns raised by the Reviewers. All points raised by the Reviewers are addressed in detail in the individual responses.

---

### Meta-Review · Area_Chair_Zo6p · 2026-01-08

**Summary:**

The paper proposes Constrained Flow Optimization (CFO), a framework utilizing Augmented Lagrangian methods to fine-tune generative flow models for reward maximization under hard constraints. The initial review status was polarized (scores: 6, 6, 2, 2). While there is consensus on the soundness of the optimization framework and the clarity of the motivation, reviewers were divided on the novelty (viewed as an incremental application of classical methods by negative reviewers). Major concerns include the insufficiency of the initial empirical evaluation (missing baselines and chemical metrics), unrealistic theoretical assumptions (perfect inner solvers), limited empirical evaluation (only synthetic 2D and one molecular task),

**Reviewer Concerns:**

**Addressed**:
- Missing Baselines (jsKV, RqDF, 5dqt): The authors added a fixed-weight Lagrangian baseline (showing sensitivity to $\lambda$), a rejection sampling baseline , and more molecular property metrics (validity, QED, Lipinski).
- Presentation & Clarity (jsKV): Section 5 was rewritten to clarify the theoretical implications and assumptions.
- Computational Cost (jsKV): Clarified that CFO and the baseline (Adjoint Matching) use the same total gradient budget, with only a ~5% runtime overhead for CFO.

**Partially Addressed**:
- Unrealistic Theoretical Assumptions (RqDF): The reviewer noted that assuming a perfect inner solver ($T \to \infty$) invalidates the practical relevance of the theory. The authors argued this is standard for oracle analyses and that coarse steps work empirically. While true, the gap between the formal guarantees and the practical algorithm remains.
- Molecular Validity (5dqt): The reviewer noted reduced chemical validity in generated molecules. The authors showed CFO generates more valid molecules than the unconstrained baseline (AM), but the absolute validity numbers (and the trade-off with the reward) confirm that the method struggles to maintain high quality while optimizing aggressive constraints.

**Outstanding**:
- Incremental Contribution (6dLh): The core critique that applying augmented Lagrangian to fine-tuning is a standard engineering application rather than a methodological breakthrough remains. The rebuttal defends this as "standard practice," but it doesn't change the fact that the novelty is primarily in the application of a classical method.
- Limited Scope (RqDF): The reviewer questioned why only molecular design was tested if the method is general. The authors renamed the paper to focus on molecular design rather than expanding the scope, effectively conceding the limited applicability.

**Reviewer Scores:**

- Reviewer 5dqt (Original: 6): Predicted: 6 (Marginally Above). The additional metrics (QED, Lipinski) provide a more complete picture, likely solidifying their "Marginal" vote, but the fundamental concern about molecular quality remains relevant.
- Reviewer jsKV (Original: 6): Predicted: 6 (Marginally Above). The fixed-weight baseline and improved writing address their specific requests. They are likely to stick with their positive (but not enthusiastic) score.
- Reviewer 6dLh (Original: 2): Predicted: 2 (Reject). This reviewer viewed the contribution as technically incremental and the baselines as insufficient. While new baselines were added, the core "novelty" critique remains.
- Reviewer RqDF (Original: 2): Predicted: 4 (Marginally Below). While the rejection sampling baseline helps, the fundamental skepticism about the theoretical assumptions ("unrealistic") and the narrow application scope likely prevent a flip to Accept.

---

### Decision · Program_Chairs · 2026-01-26

Reject